# DISCRIMINATIVE SIMILARITY FOR DATA CLUSTERING

**Yingzhen Yang**
School of Computing and Augmented Intelligence
Arizona State University
Tempe, AZ 85281, USA
yingzhen.yang@asu.edu

**Ping Li**
Cognitive Computing Lab
Baidu Research
Bellevue, WA 98004, USA
liping11@baidu.com

## ABSTRACT

Similarity-based clustering methods separate data into clusters according to the pairwise similarity between the data, and the pairwise similarity is crucial for their performance. In this paper, we propose *Clustering by Discriminative Similarity (CDS)*, a novel method which learns discriminative similarity for data clustering. CDS learns an unsupervised similarity-based classifier from each data partition, and searches for the optimal partition of the data by minimizing the generalization error of the learnt classifiers associated with the data partitions. By generalization analysis via Rademacher complexity, the generalization error bound for the unsupervised similarity-based classifier is expressed as the sum of discriminative similarity between the data from different classes. It is proved that the derived discriminative similarity can also be induced by the integrated squared error bound for kernel density classification. In order to evaluate the performance of the proposed discriminative similarity, we propose a new clustering method using a kernel as the similarity function, CDS via unsupervised kernel classification (CDSK), with its effectiveness demonstrated by experimental results.

## 1 INTRODUCTION

Similarity-based clustering methods segment the data based on the similarity measure between the data points, such as spectral clustering (Ng et al., 2001), pairwise clustering method (Shental et al., 2003), K-means (Hartigan & Wong, 1979), and kernel K-means (Schölkopf et al., 1998). The success of similarity-based clustering highly depends on the underlying pairwise similarity over the data, which in most cases are constructed empirically, e.g., by Gaussian kernel or the K-Nearest-Neighbor (KNN) graph. In this paper, we model data clustering as a multiclass classification problem and seek for the data partition where the associated classifier, trained on cluster labels, can have low generalization error. Therefore, it is natural to formulate data clustering problem as a problem of training unsupervised classifiers: a classifier can be trained upon each candidate partition of the data, and the quality of the data partition can be evaluated by the performance of the trained classifier. Such classifier trained on a hypothetical labeling associated with a data partition is termed an unsupervised classifier.

We present *Clustering by Discriminative Similarity (CDS)*, wherein discriminative similarity is derived by the generalization error bound for an unsupervised similarity-based classifier. CDS is based on a novel framework of discriminative clustering by unsupervised classification wherein an unsupervised classifier is learnt from unlabeled data and the preferred hypothetical labeling should minimize the generalization error bound for the learnt classifier. When the popular Support Vector Machines (SVMs) is used in this framework, unsupervised SVM (Xu et al., 2004) can be deduced. In this paper, a similarity-based classifier motivated by similarity learning (Balcan et al., 2008; Cortes et al., 2013), is used as the unsupervised classifier. By generalization analysis via Rademacher complexity, the generalization error bound for the unsupervised similarity-based classifier is expressed as sum of pairwise similarity between the data from different classes. Such pairwise similarity, parameterized by the weights of the unsupervised similarity-based classifier, serves as the discriminative similarity. The term "discriminative similarity" emphasizes the fact that the similarity is learnt so as

---

Yingzhen Yang's work was conducted as a consulting researcher at Baidu Research - Bellevue, WA, USA.

to improve the discriminative capability of a certain classifier such as the aforementioned unsupervised similarity-based classifier.

## 1.1 CONTRIBUTIONS AND MAIN RESULTS

Firstly, we present Clustering by Discriminative Similarity (CDS) where discriminative similarity is induced by the generalization error bound for unsupervised similarity-based classifier on unlabeled data. The generalization bound for such similarity-based classifier is of independent interest, which is among the few results of generalization bounds for classification using general similarity functions (Section B.1 of Appendix). When the general similarity function is set to a Positive Semi-Definite (PSD) kernel, the derived discriminative similarity between two data points $\mathbf{x}_i,\mathbf{x}_j$ is $S_{ij}^K = 2(\alpha_i + \alpha_j - \lambda \alpha_i \alpha_j)K(\mathbf{x}_i - \mathbf{x}_j)$, where $K$ can be an arbitrary PSD kernel and $\alpha_i$ is the kernel weight associated with $\mathbf{x}_i$. With theoretical and empirical study, we argue that $S_{ij}^K$ should be used for data clustering instead of the conventional kernel similarity corresponding to uniform kernel weights. In the case of binary classification, we prove that the derived discriminative similarity $S_{ij}^K$ has the same form as the similarity induced by the integrated squared error bound for kernel density classification (Section A of the appendix). Such connection suggests that there exists information-theoretic measure which is implicitly equivalent to our CDS framework for unsupervised learning, and our CDS framework is well grounded for learning similarity from unlabeled data.

Secondly, based on our CDS model, we develop a clustering algorithm termed Clustering by Discriminative Similarity via unsupervised Kernel classification (CDSK) in Section 5. CDSK uses a PSD kernel as the similarity function, and outperforms competing clustering algorithms, including nonparametric discriminative similarity based clustering methods and similarity graph based clustering methods, demonstrating the effectiveness of CDSK. When the kernel weights $\{\alpha_i\}$ are uniform, CDSK is equivalent to kernel K-Means (Schölkopf et al., 1998). CDSK is more flexible by learning adaptive kernel weights associated with different data points.

## 1.2 CONNECTION TO RELATED WORKS

Our CDS model is related to a class of discriminative clustering methods which classify unlabeled data by various measures on discriminative unsupervised classifiers, and the measures include generalization error (Xu et al., 2004) or the entropy of the posterior distribution of the label (Gomes et al., 2010). Discriminative clustering methods (Xu et al., 2004) predict the labels of unlabeled data by minimizing the generalization error bound for the unsupervised classifier with respect to the hypothetical labeling. Unsupervised SVM is proposed in Xu et al. (2004) which learns a binary classifier to partition unlabeled data with the maximum margin between different clusters. The theoretical properties of unsupervised SVM are further analyzed in Karnin et al. (2012). Kernel logistic regression classifier is employed in Gomes et al. (2010), and it uses the entropy of the posterior distribution of the class label by the classifier to measure the quality of the hypothetical labeling. CDS model performs discriminative clustering based on a novel unsupervised classification framework by considering similarity-based or kernel classifiers which are important classification methods in the supervised learning literature. In contrasts with kernel similarity with uniform weights, the induced discriminative similarity with learnable weights enhances its capability to represent complex interconnection between data. The generalization analysis for CDS is primarily based on distribution free Rademacher complexity. While Yang et al. (2014a) propose nonparametric discriminative similarity for clustering, the nonparametric similarity requires probability density estimation which is difficult for high-dimensional data, and the fixed nonparametric similarity is not adaptive to complicated data distribution.

The paper is organized as follows. We introduce the problem setup of Clustering by Discriminative Similarity in Section 3. We then derive the generalization error bound for the unsupervised similarity-based classifier for CDS in Section 4 where the proposed discriminative similarity is induced by the error bound. The application of CDS to data clustering is shown in Section 5. Throughout this paper, the term kernel stands for the PSD kernel if no special notes are made.

## 2 SIGNIFICANCE OF CDSK OVER EXISTING DISCRIMINATIVE AND SIMILARITY-BASED CLUSTERING METHODS

Effective data similarity highly depends on the underlying probabilistic distribution and geometric structure of the data, and these two characteristics leads to "data-driven" similarity, such as Zhu et al. (2014); Bicego et al. (2021); Ng et al. (2001); Shental et al. (2003); Hartigan & Wong (1979); Schölkopf et al. (1998) and similarity based on geometric structure of the data, such as the subspace structure (Sparse Subspace Clustering, or SSC in Elhamifar & Vidal (2013)). Note that the sparse graph method, $\ell^1$-Graph (Yan & Wang, 2009), has the same formulation as SSC. Most existing clustering methods based on data-driven or geometric structure-driven similarity suffer from a common deficiency, that is, the similarity is not explicitly optimized for the purpose of separating underlying clusters. In particular, the Random Forest-based similarity (Zhu et al., 2014; Bicego et al., 2021) is extracted from features in decision trees. Previous works about subspace-based similarity (Yan & Wang, 2009; Elhamifar & Vidal, 2013) try to make sure that only data points lying on or close to the same subspace have nonzero similarity, so that data points from the same subspace can form a cluster. However, it is not guaranteed that features in the decision trees are discriminative enough to separate clusters, because the candidate data partition (or candidate cluster labels) do not participate in the feature or similarity extraction process. Note that synthetically generated negative class are suggested in Zhu et al. (2014); Bicego et al. (2021) to train unsupervised random forest, however, the synthetic labels are not for the original data. Moreover, it is well known that the existing subspace learning methods only obtain reliable subspace-based similarity with restrictive geometric assumptions on the data and the underlying subspaces, such as large principal angle between intersecting subspaces (Soltanolkotabi & Candes, 2012; Elhamifar & Vidal, 2013).

Therefore, it is particularly important to derive similarity for clustering which meets two requirements: (1) discriminative measure with information such as cluster partition is used to derive such similarity so as to achieve compelling clustering performance; (2) it requires less restrictive assumptions on the geometric structure of the data than current geometric structure-based similarity learning methods, such as subspace clustering (Yan & Wang, 2009; Elhamifar & Vidal, 2013).

**Significance.** The proposed discriminative similarity of this paper meets these two requirements. First, the discriminative similarity is derived by the generalization error bound associated with candidate cluster labeling, and minimizing the objective function of our optimization problem for clustering renders a joint optimization of discriminative similarity and candidate cluster labeling in a way such that the similarity-based classifier has small generalization error bound. Second, our framework only assumes a mild classification model in Definition 3.1, which only requires an unknown joint distribution over data and its labels. In this way, the restrictive geometric assumptions are avoided in our method. Compared to the existing discriminative clustering methods, such as MMC (Xu et al., 2004), BMMC (Chen et al., 2014), RIM (Gomes et al., 2010), and the other discriminative clustering methods such as (Huang et al., 2015; Nguyen et al., 2017), the optimization problem of CDSK with discriminative similarity-based formulation is much easier to solve and it enjoys convexity and efficiency in each iteration of coordinate descent described in Algorithm 1. In particular, as mentioned in Section D of the appendix, the first step (11) of each iteration can be solved by efficient SVD or other randomized large-scale SVD methods, and the second step (12) of each iteration can be solved by efficient SMO (Platt, 1998). Moreover, the optimization problems in these two steps are either convex or having closed-form solution. In contrast, MMC requires expensive semidefinite programming. RIM has to solve a nonconvex optimization problem and its formulation does not guarantee that the trained multi-class kernelized logistic regression has low classification error on candidate labeling, which explains why it has inferior performance compared to our method. The discriminative Extreme Learning Machine (Huang et al., 2015) trains ELM using labels produced by a simple clustering method such as K-means, and the potentially poor cluster labels by the simple clustering method can easily result in unsatisfactory performance of this method. The discriminative Bayesian nonparametric clustering (Nguyen et al., 2017) and BMMC (Chen et al., 2014) require extra efforts of sampling hidden variables and tuning hyperparameters to generate the desirable number of clusters (or model selection), which could reduce the effect of discriminative measures used in these Bayesian nonparametric methods.

## 3 PROBLEM SETUP

We introduce the problem setup of the formulation of clustering by unsupervised classification. Given unlabeled data $\{\mathbf{x}_l\}_{l=1}^n \subset \mathbb{R}^d$, clustering is equivalent to searching for the hypothetical labeling which is optimal in some sense. Each hypothetical labeling corresponds to a candidate data partition. Figure 1 illustrates four binary hypothetical labelings which correspond to four partitions of the data, and the data is divided into two clusters by each hypothetical labeling.

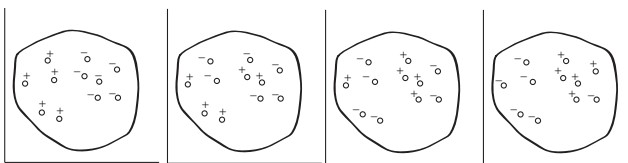

Figure 1: Illustration of binary hypothetical labelings

The discriminative clustering literature (Xu et al., 2004; Gomes et al., 2010) has demonstrated the potential of multi-class classification for clustering problem. Inspired by the natural connection between clustering and classification, we proposes the framework of Clustering by Unsupervised Classification which models clustering problem as a multi-class classification problem. A classifier is learnt from unlabeled data with a hypothetical labeling, which is associated with a candidate partition of the unlabeled data. The optimal hypothetical labeling is supposed to be the one such that its associated classifier has the minimum generalization error bound. To study the generalization bound for the classifier learnt from hypothetical labeling, the concept of classification model is needed. Given unlabeled data $\{\mathbf{x}_l\}_{l=1}^n$, a classification model $M_{\mathcal{Y}}$ is constructed for any hypothetical labeling $\mathcal{Y} = \{y_l\}_{l=1}^n$ as follows.

**Definition 3.1.** The classification model corresponding to the hypothetical labeling $\mathcal{Y} = \{y_l\}_{l=1}^n$ is defined as $M_{\mathcal{Y}} = (\mathcal{S}, F)$. $\mathcal{S} = \{\mathbf{x}_l, y_l\}_{l=1}^n$ are the labeled data by the hypothetical labeling $\mathcal{Y}$, and $\mathcal{S}$ are assumed to be i.i.d. samples drawn from the some unknown joint distribution $P_{XY}$, where $(X, Y)$ is a random couple, $X \in \mathcal{X} \subseteq \mathbb{R}^d$ represents the data in some compact domain $\mathcal{X}$, and $Y \in \{1, 2, ..., c\}$ is the class label of $X$, $c$ is the number of classes. $F$ is a classifier trained on $\mathcal{S}$. The generalization error of the classification model $M_{\mathcal{Y}}$ is defined as the generalization error of the classifier $F$ in $M_{\mathcal{Y}}$.

**The basic assumption of CDS is that the optimal hypothetical labeling minimizes the generalization error bound for the classification model.** With $f$ being different classifiers, different discriminative clustering models can be derived. When SVMs is used as the classifier $F$ in the above discriminative model, unsupervised SVM (Xu et al., 2004) is obtained.

In Balcan et al. (2008), the authors proposes a classification method using general similarity functions. The classification rule measures the similarity of the test data to each class, and then assigns the test data to the class such that the weighed average of the similarity between the test data and the training data belonging to that class is maximized over all the classes. Inspired by this classification method, we now consider using a general symmetric and continuous function $S\colon \mathcal{X} \times \mathcal{X} \to [0, 1]$ as the similarity function in our CDS model. We propose the following hypothesis,

$$h_S(\mathbf{x}, y) = \sum_{i\colon y_i = y} \alpha_i S(\mathbf{x}, \mathbf{x}_i). \tag{1}$$

In the next section, we derive generalization bound for the unsupervised similarity-based classifier based on the above hypothesis, and such generalization bound leads to discriminative similarities for data clustering. When $S$ is a PSD kernel, minimizing the generalization error bound amounts to minimization of a new form of kernel similarity between data from different clusters, which lays the foundation of a new clustering algorithm presented in Section 5.

## 4 GENERALIZATION BOUND FOR SIMILARITY-BASED CLASSIFIER

In this section, the generalization error bound for the classification model in Definition 3.1 with the unsupervised similarity-based classifier is derived as a sum of discriminative similarity between the data from different classes.

### 4.1 GENERALIZATION BOUND

The following notations are introduced before our analysis. Let $\boldsymbol{\alpha} = [\alpha_1, \ldots, \alpha_n]^\top$ be the nonzero weights that sum up to 1, $\boldsymbol{\alpha}^{(y)}$ be a $n \times 1$ column vector representing the weights belonging to class $y$ such that $\alpha_i^{(y)}$ is $\alpha_i$ if $y = y_i$, and 0 otherwise. The margin of the labeled sample $(\mathbf{x}, y)$ is defined as $m_{h_S}(\mathbf{x}, y) = h_S(\mathbf{x}, y) - \mathrm{argmax}_{y' \neq y} h_S(\mathbf{x}, y')$, the sample $(\mathbf{x}, y)$ is classified correctly if $m_{h_S}(\mathbf{x}, y) \geq 0$.

The general similarity-based classifier $f_S$ predicts the label of the input $\mathbf{x}$ by $f_S(\mathbf{x}) = \mathrm{argmax}_{y \in \{1, \ldots, c\}} h_S(\mathbf{x}, y)$. We then begin to derive the generalization error bound for $f_S$ using the Rademacher complexity of the function class comprised of all the possible margin functions $m_{h_S}$. The Rademacher complexity (Bartlett & Mendelson, 2003; Koltchinskii, 2001) of a function class is defined below:

**Definition 4.1.** Let $\{\sigma_i\}_{i=1}^n$ be $n$ i.i.d. random variables such that $\Pr[\sigma_i = 1] = \Pr[\sigma_i = -1] = \frac{1}{2}$. The Rademacher complexity of a function class $\mathcal{A}$ is defined as

$$\mathfrak{R}(\mathcal{A}) = \mathbb{E}_{\{\sigma_i\}, \{\mathbf{x}_i\}} \left[ \sup_{h \in \mathcal{A}} \left| \frac{1}{n} \sum_{i=1}^n \sigma_i h(\mathbf{x}_i) \right| \right]. \tag{2}$$

In order to analyze the generalization property of the classification rule using the general similarity function, we first investigate the properties of general similarity function and its relationship to PSD kernels in terms of eigenvalues and eigenfunctions of the associated integral operator. The integral operator $(L_S f)(\mathbf{x}) = \int S(\mathbf{x}, \mathbf{t}) f(\mathbf{t}) d\mathbf{t}$ is well defined. It can be verified that $L_S$ is a compact operator since $S$ is continuous. According to the spectral theorem in operator theory, there exists an orthonormal basis $\{\phi_1, \phi_2, \ldots\}$ of $\mathcal{L}^2$ which is comprised of the eigenfunctions of $L_S$, where $\mathcal{L}^2$ is the space of measurable functions which are defined over $\mathcal{X}$ and square Lebesgue integrable. $\phi_k$ is the eigenfunction of $L_S$ with eigenvalue $\lambda_k$ if $L_S \phi_k = \lambda_k \phi_k$. The following lemma shows that under certain assumption on the eigenvalues and eigenfunctions of $L_S$, a general symmetric and continuous similarity can be decomposed into two PSD kernels.

**Lemma 4.1.** Suppose $S \colon \mathcal{X} \times \mathcal{X} \to [0, 1]$ is a symmetric continuous function, and $\{\lambda_k\}$ and $\{\phi_k\}$ are the eigenvalues and eigenfunctions of $L_S$ respectively. Suppose $\sum_{k \geq 1} \lambda_k |\phi_k(\mathbf{x})|^2 < C$ for some constant $C > 0$. Then $S(\mathbf{x}, \mathbf{t}) = \sum_{k \geq 1} \lambda_k \phi_k(\mathbf{x}) \phi_k(\mathbf{t})$ for any $\mathbf{x}, \mathbf{t} \in \mathcal{X}$, and it can be decomposed as the difference between two positive semi-definite kernels: $S(\mathbf{x}, \mathbf{t}) = S^+(\mathbf{x}, \mathbf{t}) - S^-(\mathbf{x}, \mathbf{t})$, with

$$S^+(\mathbf{x}, \mathbf{t}) = \sum_{k \colon \lambda_k \geq 0} \lambda_k \phi_k(\mathbf{x}) \phi_k(\mathbf{t}), \quad S^-(\mathbf{x}, \mathbf{t}) = \sum_{k \colon \lambda_k < 0} |\lambda_k| \phi_k(\mathbf{x}) \phi_k(\mathbf{t}). \tag{3}$$

We now use a regularization term to bound the Rademacher complexity for the classification rule using the general similarity function. Let $\Omega^+(\boldsymbol{\alpha}) = \sum_{y=1}^c \boldsymbol{\alpha}^{(y)^\top} \mathbf{S}^+ \boldsymbol{\alpha}^{(y)}$ and $\Omega^-(\boldsymbol{\alpha}) = \sum_{y=1}^c \boldsymbol{\alpha}^{(y)^\top} \mathbf{S}^- \boldsymbol{\alpha}^{(y)}$ with $[\mathbf{S}^+]_{ij} = S^+(\mathbf{x}_i, \mathbf{x}_j)$ and $[\mathbf{S}^-]_{ij} = S^-(\mathbf{x}_i, \mathbf{x}_j)$. The space $\mathcal{H}_y$ of all the hypothesis $h_S(\cdot, y)$ associated with label $y$ is defined as

$$\mathcal{H}_{S,y} = \left\{ (\mathbf{x}, y) \to \sum_{i \colon y_i = y} \alpha_i S(\mathbf{x}, \mathbf{x}_i) \colon \boldsymbol{\alpha} \geq \mathbf{0}, \mathbf{1}^\top \boldsymbol{\alpha} = 1, \Omega^+(\boldsymbol{\alpha}) \leq B^{+2}, \Omega^-(\boldsymbol{\alpha}) \leq B^{-2} \right\}$$

for $1 \leq y \leq c$, with positive number $B^+$ and $B^-$ which bound $\Omega^+$ and $\Omega^-$ respectively. Let the hypothesis space comprising all possible margin functions be $\mathcal{H}_S = \{(\mathbf{x}, y) \to m_{h_S}(\mathbf{x}, y) \colon h_S(\mathbf{x}, y) \in \mathcal{H}_{S,y}\}$. We then present the main result in this section about the generalization error of unsupervised similarity-based classifier $f_S$.

**Theorem 4.2.** Given the discriminative model $M_{\mathcal{Y}} = (\mathcal{S}, f_S)$, suppose $\Omega^+(\boldsymbol{\alpha}) \leq B^{+2}, \Omega^-(\boldsymbol{\alpha}) \leq B^{-2}, \sup_{\mathbf{x} \in \mathcal{X}} |S^+(\mathbf{x}, \mathbf{x})| \leq R^2, \sup_{\mathbf{x} \in \mathcal{X}} |S^-(\mathbf{x}, \mathbf{x})| \leq R^2$ for positive constants $B^+, B^-$ and $R$. Then with probability $1 - \delta$ over the labeled data $\mathcal{S}$ with respect to any distribution in $P_{XY}$, under

the assumptions of Lemma 4.1, the generalization error of the general classifier $f_S$ satisfies

$$R(f_S) = \Pr\left[Y \neq f_S(X)\right]$$

$$\leq \widehat{R}_n(f_S) + \frac{8R(2c-1)c(B^+ + B^-)}{\gamma\sqrt{n}} + \left(\frac{16c(2c-1)(B^+ + B^-)R^2}{\gamma} + 1\right)\sqrt{\frac{\log\frac{4}{\delta}}{2n}}, \tag{4}$$

where $\widehat{R}_n(f_S) = \frac{1}{n}\sum_{i=1}^{n}\Phi\left(\frac{h_S(\mathbf{x}_i, y_i) - \mathrm{argmax}_{y' \neq y}h_S(\mathbf{x}_i, y')}{\gamma}\right)$ is the empirical loss of $f_S$ on the labeled data, $\gamma > 0$ is a constant and $\Phi$ is defined as $\Phi(x) = \min\{1, \max\{0, 1-x\}\}$. Moreover, if $\gamma \geq 1$, the empirical loss $\widehat{R}_n(f_S)$ satisfies

$$\widehat{R}_n(f_S) \leq 1 - \frac{1}{n\gamma}\sum_{i,j=1}^{n}\frac{\alpha_i + \alpha_j}{2}S(\mathbf{x}_i, \mathbf{x}_j) + \frac{1}{n\gamma}\sum_{1 \leq i < j \leq n}2(\alpha_i + \alpha_j)S(\mathbf{x}_i, \mathbf{x}_j)\mathbb{I}_{y_i \neq y_j}. \tag{5}$$

The indicator function $\mathbb{I}_E$ in (5) is 1 if event $E$ is true, and 0 otherwise.

**Remark 4.3.** Lemma E.3 in the Appendix shows that the Rademacher complexity of $\mathcal{H}_S$ is bounded in terms of $B^+$ and $B^-$, and that is why these two quantities appear on the RHS of (4). In addition, when $S$ is a Positive Semi-Definite (PSD) kernel $K$, it can be verified that $S^- \equiv 0$, $S = S^+$.

**Remark 4.4.** When the decomposition $S = S^+ - S^-$ exists and $S^+$, $S^-$ are PSD kernels, $S$ is the kernel of some Reproducing Kernel Kreĭn Space (RKKS) (Mary, 2003). Ong et al. (2004) and Loosli et al. (2016) analyzed the problem of learning SVM-style classifiers with indefinite kernels from the Kreĭn space. However, their work does not show when and how an indefinite and general similarity function can have PSD decomposition, as well as the generalization analysis for the similarity-based classifier using such general indefinite function as similarity measure. Our analysis deals with these problems in Lemma 4.1 and Theorem 4.2. It should be emphasized that our generalization bound is of independent interest in supervised learning, because it is among the few results of generalization bounds using general similarity-based classifier. Section B.1 shows that the our bound is a principled result with strong connection to established generalization error bound for Support Vector Machines (SVMs) or Kernel Machines.

## 4.2 Clustering by Discriminative Similarity

We let

$$S_{ij}^{\text{sim}} = 2(\alpha_i + \alpha_j)S(\mathbf{x}_i, \mathbf{x}_j) - 2\lambda\alpha_i\alpha_j S^+(\mathbf{x}_i, \mathbf{x}_j) - 2\lambda\alpha_i\alpha_j S^-(\mathbf{x}_i, \mathbf{x}_j) \tag{6}$$

be the discriminative similarity between data from different classes, which is induced by the generalization error bound (4) for the unsupervised general similarity-based classifier $f_S$. Minimizing the bound (4) motivates us to consider the optimization problem that minimizes $\widehat{R}_n(f_S) + \lambda\left(\Omega^+(\boldsymbol{\alpha}) + \Omega^-(\boldsymbol{\alpha})\right)$. Replacing $\widehat{R}_n(f_S)$ by its upper bound in (5), we consider the following problem,

$$\min_{\boldsymbol{\alpha}, \mathcal{Y}} \sum_{1 \leq i < j \leq n} S_{ij}^{\text{sim}}\mathbb{I}_{y_i \neq y_j} - \sum_{i,j=1}^{n}\frac{\alpha_i + \alpha_j}{2}S(\mathbf{x}_i, \mathbf{x}_j) + \lambda(\boldsymbol{\alpha}^\top \mathbf{S}^+ \boldsymbol{\alpha} + \boldsymbol{\alpha}^\top \mathbf{S}^- \boldsymbol{\alpha})$$

$$s.t. \ \boldsymbol{\alpha} \geq \mathbf{0}, \mathbf{1}^\top \boldsymbol{\alpha} = 1, \mathcal{Y} = \{y_i\}_{i=1}^{n}, \tag{7}$$

where $\lambda > 0$ is the weighting parameter for the regularization term $\Omega^+(\boldsymbol{\alpha}) + \Omega^-(\boldsymbol{\alpha})$. Note that we do not set $\lambda$ to $\frac{16c(2c-1)R^2 + 8\sqrt{2}R(2c-1)c}{\sqrt{2}\gamma}$ exactly matching the RHS of (4), because $\lambda$ controls the weight of the regularization term which bounds the unknown complexity of the function class $\mathcal{H}_S$. Note that (7) encourages the discriminative similarity $S_{ij}^{\text{sim}}$ between the data from different classes small. The optimization problem (7) forms the formulation of Clustering by Discriminative Similarity (CDS).

By Remark 4.3, when $S$ is a PSD kernel $K$, $S^- \equiv 0$, $S = S^+$, $S_{ij}^{\text{sim}}$ reduces to the following discriminative similarity for PSD kernels:

$$S_{ij}^K = 2(\alpha_i + \alpha_j - \lambda\alpha_i\alpha_j)K(\mathbf{x}_i - \mathbf{x}_j), 1 \leq i, j \leq n, \tag{8}$$

and $S_{ij}^K$ is the similarity induced by the unsupervised kernel classifier by the kernel $K$.

Without loss of generality, we set $K = K_\tau(\mathbf{x}) = \exp(-\frac{\|\mathbf{x}\|_2^2}{2\tau^2})$ which is the isotropic Gaussian kernel with kernel bandwidth $\tau > 0$, and we omit the constant that makes integral of $K$ unit.

When setting the general similarity function to kernel $K_\tau$, CDS aims to minimize the error bound for the corresponding unsupervised kernel classifier, which amounts to minimizing the following objective function

$$\min_{\boldsymbol{\alpha} \in \Lambda, \mathcal{Y} = \{y_i\}_{i=1}^n} \sum_{1 \leq i < j \leq n}^n S_{ij}^K \mathbb{1}_{y_i \neq y_j} - \sum_{i,j=1}^n \frac{\alpha_i + \alpha_j}{2} K_\tau(\mathbf{x}_i - \mathbf{x}_j) + \lambda \boldsymbol{\alpha}^\top \mathbf{K} \boldsymbol{\alpha}, \qquad (9)$$

where $S_{ij}^K$ is defined in (8) with $K = K_\tau$. $\mathbf{K} \in \mathbb{R}^{n \times n}$ and $\mathbf{K}_{ij} = K_\tau(\mathbf{x}_i - \mathbf{x}_j)$. $\lambda$ is tuned such that $S_{ij}^K \geq 0$, e.g., $\lambda \leq 2$. In Section A, it is shown that the discriminative similarity (8) can also be induced from the perspective of kernel density classification by kernel density estimators with nonuniform weights. It supports the theoretical justification for the induced discriminative similarity in this section.

## 5 Application to Data Clustering

In this section, we propose a novel data clustering method termed Clustering by Discriminative Similarity via unsupervised Kernel classification (CDSK) which is an empirical method inspired by our CDS model when the similarity function is a PSD kernel $K = K_\tau$. In accordance with the CDS model in Section 4.2, CDSK aims to minimize (9). However, problem (9) involves minimization with respect to discrete cluster labels $\mathcal{Y} = \{y_i\}$ which is NP-hard. In addition, it potentially results in a trivial solution which puts all the data in a single cluster due to the lack of constraints on the cluster balance. When $\mathbf{Y}$ is a binary matrix where each column is a membership vector for a particular cluster, $\sum_{1 \leq i < j \leq n}^n S_{ij}^K \mathbb{1}_{y_i \neq y_j} = \frac{1}{2} \text{Tr}(\mathbf{Y}^\top \mathbf{L}^K \mathbf{Y})$. Therefore, (9) is relaxed in the proposed optimization problem for CDSK below:

$$\min_{\boldsymbol{\alpha} \in \Lambda, \mathbf{Y} \in \mathbb{R}^{n \times c}} \frac{1}{2} \text{Tr}(\mathbf{Y}^\top \mathbf{L}^K \mathbf{Y}) - \sum_{i,j=1}^n \frac{\alpha_i + \alpha_j}{2} K_\tau(\mathbf{x}_i - \mathbf{x}_j) + \lambda \boldsymbol{\alpha}^\top \mathbf{K} \boldsymbol{\alpha} \quad s.t. \ \mathbf{Y}^\top \mathbf{D}^K \mathbf{Y} = \mathbf{I}_c,$$

$$(10)$$

where $\Lambda = \{\boldsymbol{\alpha} \colon \boldsymbol{\alpha} \geq \mathbf{0}, \mathbf{1}^\top \boldsymbol{\alpha} = 1\}$, $\mathbf{S}_{ij}^K = S_{ij}^K$, $\mathbf{L}^K = \mathbf{D}^K - \mathbf{S}^K$ is the graph Laplacian computed with $\mathbf{S}^K$, $\mathbf{D}^K$ is a diagonal matrix with each diagonal element being the sum of the corresponding row of $\mathbf{S}^K$: $[\mathbf{D}^K]_{ii} = \sum_{j=1}^n \mathbf{S}_{ij}^K$, $\mathbf{I}_c$ is a $c \times c$ identity matrix, $c$ is the number of clusters. The constraint in (10) is used to balance the cluster size. This is because minimizing (9) without any constraint on the cluster size results in a trivial solution where all data points form a single cluster. Inspired by spectral clustering (Ng et al., 2001), the constraint $\mathbf{Y}^\top \mathbf{D}^K \mathbf{Y} = \mathbf{I}_c$ used in CDSK prevents imbalanced data clusters.

Problem (10) is optimized by coordinate descent. In each iteration of coordinate descent, optimization with respect to $\mathbf{Y}$ is performed with fixed $\boldsymbol{\alpha}$, which is exactly the same problem as that of spectral clustering with a solution formed by the smallest $c$ eigenvectors of the normalized graph Laplacian $(\mathbf{D}^K)^{-1/2} \mathbf{L}^K (\mathbf{D}^K)^{-1/2}$; then the optimization with respect to $\boldsymbol{\alpha}$ is performed with fixed $\mathbf{Y}$, which is a standard constrained quadratic programming problem. The iteration of coordinate descent proceeds until convergence or the maximum iteration number $M$ is achieved. Each iteration solves two subproblems, (11) and (12). In order to promote sparsity of $\boldsymbol{\alpha}$, $\boldsymbol{\alpha}$ can be initialized by solving $\sum_{i=1}^n \left\| \mathbf{x}_i - \sum_{j \neq i} \mathbf{x}_j \alpha_j \right\|_2^2 + \tau \|\boldsymbol{\alpha}\|_0$ for a positive weighting parameter $\tau = 0.1$. The algorithm of CDSK is described in Algorithm 1.

Furthermore, Section C in the appendix explains the theoretical properties of the coordinate descent algorithm for problem (10).

The baseline named SC-NS performs spectral clustering on the nonparametric similarity proposed in Yang et al. (2014a). The baseline named SC-MS first constructs a similarity matrix between data

---

**Algorithm 1** Clustering by Discriminative Similarity via unsupervised Kernel classification (CDSK)

---

Input: Unlabeled dataset $\{\mathbf{x}_l\}_{l=1}^n$, parameter $\lambda$, maximum iteration number $M$.
**for** $t \leftarrow 1$ to $M$ **do**
    With fixed $\boldsymbol{\alpha}$, solve

$$\min_{\mathbf{Y} \in \mathbb{R}^{n \times c}} \text{Tr}(\mathbf{Y}^\top \mathbf{L}^K \mathbf{Y}) \quad s.t. \ \mathbf{Y}^\top \mathbf{D}^K \mathbf{Y} = \mathbf{I}_c, \tag{11}$$

With fixed $\mathbf{Y}$, solve

$$\min_{\boldsymbol{\alpha} \in \Lambda,} \text{Tr}(\mathbf{Y}^\top \mathbf{L}^K \mathbf{Y}) - \sum_{i,j=1}^n \frac{\alpha_i + \alpha_j}{2} K_\tau(\mathbf{x}_i - \mathbf{x}_j) + \lambda \boldsymbol{\alpha}^\top \mathbf{K} \boldsymbol{\alpha}$$

$$s.t. \ \mathbf{Y}^\top \mathbf{D}^K \mathbf{Y} = \mathbf{I}_c, \tag{12}$$

**end for**
Perform K-Means Clustering on rows of $\mathbf{Y}$ to obtain the clustering result.

---

denoted by $\mathbf{W}$, where $\mathbf{W}_{ij} = K_\tau(\mathbf{x}_i - \mathbf{x}_j)$, then optimize the kernel bandwidth $h$ by minimizing $\sum_i \|\mathbf{x}_i - \frac{1}{d_i} \sum_j \mathbf{W}_{ij} \mathbf{x}_j\|_2$ where $d_i = \sum_j \mathbf{W}_{ij}$. SC-MS then performs spectral clustering on $\mathbf{W}$ with the kernel bandwidth $h$ obtained from the optimization.

To demonstrate the advantage of the proposed parametric discriminative similarity, we compare CDSK to various baseline clustering methods. SC stands for Spectral Clustering, which is the best performer among spectral clustering with similarity matrix set by Gaussian kernel (SCK), spectral clustering with similarity matrix set by a manifold-based similarity learning method (SC-MS) (Karasuyama & Mamitsuka, 2013), and spectral clustering with similarity matrix set by the nonparametric discriminative similarity (SC-NS) in Yang et al. (2014a). In SC-MS, Gaussian kernel is used as data similarity, and the parameters of the diagonal covariance matrix is optimized so as to minimize the data reconstruction error term. SC-NS minimizes nonparametric kernel similarity between data across different clusters, which is the same objective as that of kernel K-Means (Schölkopf et al., 1998), so its performance is the same as kernel K-Means. Please refer to Section 2 for discussion about other baselines.

**Datasets.** We conduct experiments on the Yale face dataset, UCI Ionosphere dataset, the MNIST handwritten digits dataset and the Georgia Face dataset. The Yale face dataset has face images of 15 people with 11 images for each person. The Ionosphere data contains 351 points of dimensionality 34. The Georgia Face dataset contains images of 50 people, and each person is represented by 15 color images with cluttered background. COIL-20 dataset has 1440 images of size $32 \times 32$ for 20 objects with background removed in all images. The COIL-100 dataset contains 100 objects with 72 images of size $32 \times 32$ for each object. CMU PIE face data contains 11554 cropped face images of size $32 \times 32$ for 68 persons, and there are around 170 facial images for each person under different illumination and expressions. The UMIST face dataset is comprised of 575 images of size $112 \times 92$ for 20 people. CMU Multi-PIE (MPIE) data (Gross et al., 2010) contains 8916 facial images captured in four sessions. The MNIST handwritten digits database has a total number of 70000 samples of dimensionality 1024 for digits from 0 to 9. The digits are normalized and centered in a fixed-size image. The Extended Yale Face Database B (Yale-B) dataset contains face images for 38 subjects with 64 frontal face images taken under different illuminations for each subject. CIFAR-10 dataset consists of 50000 training images and 10000 testing images in 10 classes, and each image is a color one of size $32 \times 32$, and we perform data clustering using all the training and testing images. We also use the miniImageNet dataset used in Vinyals et al. (2016) to evaluate the potential of clustering methods. MiniImageNet consists of $60,000$ color images of size $84 \times 84$ with 100 classes, and each class has 600 images. MiniImageNet is known to be more complex than the CIFAR-10 dataset, and we perform clustering on the 64 classes in miniImageNet which are used for few-shot learning, so $38,400$ images are used for clustering. For every clustering method involving randomness such as K-Means, we report the average performance of running it for 10 times.

**Performance Measures and Tuning $\lambda$ by Cross Validation.** We use Accuracy (AC) and Normalized Mutual Information (NMI) (Zheng et al., 2004) as the performance measures. The results of different clustering methods are shown in Table 1 and Table 2 in the format of AC(NMI). Except

for SC-MS, the kernel bandwidth in all methods is set as the variance of the pairwise Euclidean distance between the data. $\lambda$ is the weight for the regularization term in our derived generalization bound. As explained in Section B.1 of the appendix, $\lambda$ plays the same role as the weight in the regularization term of SVMs or Kernel Machines. Following the common practice in the literature of SVM or Kernel Machines, $\lambda$ can be tuned by Cross-Validation (CV). While this is an unsupervised learning task and these is no labeled data for CV, we still developed a well-motivated CV procedure. Following the practice in Mairal et al. (2012), we randomly sampled $10\%$ of the given data as the validation data, then perform CDSK on the validation data. The best $\lambda$ is chosen among the discrete values between $[0.05, 05]$ with a step of $0.05$ which minimizes the average entropy of the obtained embedding matrix $\mathbf{Y} \in \mathbb{R}^{n \times c}$ by Algorithm 1, where the average entropy is compute as $\frac{1}{n} \sum_{i=1}^{n} \text{entropy}(\text{softmax}(\mathbf{Y}^i))$. This is because we would like to choose $\lambda$ which renders the most confident clustering embedding. We perform CDSK on all the datasets using this tuning strategy and observe improved performance as shown in the above two tables. For clustering methods involving random operations, the average performance over 10 runs is reported.

**Computational Complexity.** Suppose the optimization of CDSK comprises $M$ iterations of coordinate descent. The first subproblem (11) in Algorithm 1 takes $\mathcal{O}(n^2 c)$ steps using truncated Singular Value Decomposition (SVD) by Krylov subspace iterative method. We adopt Sequential Minimal Optimization (SMO) (Platt, 1998) to solve the second subproblem (12), which takes roughly $\mathcal{O}(n^{2.1})$ steps as reported in Platt (1998). Therefore, the overall time complexity of CDSK is $\mathcal{O}(Mcn^2 + Mn^{2.1})$. $M$ is set to 20 throughout all the experiments.

Table 1: Clustering results on Yale-B, Ionosphere, Georgia Face, COIL-20, COIL-100, CMU PIE and UMIST Face.

| Methods \ Dataset | Yale-B | Ionosphere | Georgia Face | COIL-20 | COIL-100 | CMU PIE | UMIST Face |
|---|---|---|---|---|---|---|---|
| K-Means | 0.09(0.13) | 0.71(0.13) | 0.50(0.69) | 0.65(0.76) | 0.49(0.75) | 0.08(0.19) | 0.42(0.64) |
| SC | 0.11(0.15) | 0.74(0.22) | 0.52(0.70) | 0.43(0.62) | 0.28(0.59) | 0.07(0.18) | 0.42(0.61) |
| $\ell^1$-Graph (Elhamifar & Vidal, 2013) | 0.79(0.78) | 0.51(0.12) | 0.54(0.70) | 0.79(0.91) | 0.53(0.80) | 0.23(0.34) | 0.44(0.65) |
| SMCE (Elhamifar & Vidal, 2011) | 0.34(0.39) | 0.68(0.09) | **0.60(0.74)** | 0.88(0.88) | 0.56(0.81) | 0.16(0.34) | 0.45(0.66) |
| Lap-$\ell^1$-Graph (Yang et al., 2014b) | 0.79(0.78) | 0.50(0.09) | 0.58(0.73) | 0.79(0.91) | 0.56(0.81) | 0.30(**0.51**) | 0.50(0.69) |
| RAG (Zhu et al., 2014) | 0.13(0.19) | 0.70(0.11) | 0.17(0.38) | 0.50(0.64) | 0.58(0.81) | 0.14(0.34) | 0.26(0.28) |
| MMC (Xu et al., 2004) | 0.71(0.69) | 0.75(0.21) | 0.42(0.58) | 0.80(0.89) | 0.61(0.63) | 0.22(0.30) | 0.51(0.56) |
| BMMC (Chen et al., 2014) | 0.65(0.63) | 0.70(0.15) | 0.34(0.41) | 0.82(0.93) | 0.64(0.69) | 0.18(0.23) | 0.55(0.61) |
| RIM (Gomes et al., 2010) | 0.62(0.74) | 0.59(0.08) | 0.39(0.56) | 0.77(0.82) | 0.71(0.79) | 0.26(0.34) | 0.40(0.53) |
| RatioRF (Bicego et al., 2021) | 0.39(0.53) | 0.62(0.05) | 0.18(0.40) | 0.65(0.75) | 0.36(0.64) | 0.15(0.36) | 0.29(0.34) |
| CDSK (**Ours**) | **0.83(0.86)** | **0.76(0.25)** | **0.60(0.74)** | **0.93(0.97)** | **0.78(0.92)** | **0.32**(0.50) | **0.67(0.80)** |

Table 2: Clustering results on CMU Multi-PIE which contains the facial images captured in four sessions (S1 to S4), MNIST, CIFAR-10, and Mini-ImageNet

| Methods \ Dataset | MPIE S1 | MPIE S2 | MPIE S3 | MPIE S4 | MNIST | CIFAR-10 | Mini-ImageNet |
|---|---|---|---|---|---|---|---|
| KM | 0.12(0.50) | 0.13(0.48) | 0.13(0.48) | 0.13(0.49) | 0.52(0.47) | 0.19(0.06) | 0.27(0.33) |
| SC | 0.13(0.53) | 0.14(0.52) | 0.14(0.52) | 0.15(0.53) | 0.38(0.36) | 0.21(0.04) | 0.29(0.35) |
| $\ell^1$-Graph (Elhamifar & Vidal, 2013) | 0.59(0.77) | 0.70(0.81) | 0.63(0.79) | 0.68(0.81) | 0.57(0.61) | 0.28(0.24) | 0.28(0.37) |
| SMCE (Elhamifar & Vidal, 2011) | 0.17(0.55) | 0.19(0.53) | 0.19(0.52) | 0.18(0.53) | 0.65(0.67) | 0.31(0.30) | 0.29(0.37) |
| Lap-$\ell^1$-Graph (Yang et al., 2014b) | 0.59(0.77) | 0.70(0.81) | 0.63(0.79) | 0.68(0.81) | 0.56(0.60) | 0.29(0.30) | 0.29(0.37) |
| RAG (Zhu et al., 2014) | 0.34(0.75) | 0.30(0.69) | 0.31(0.68) | 0.29(0.70) | 0.59(0.51) | 0.22(0.10) | 0.18(0.33) |
| MMC (Xu et al., 2004) | 0.49(0.58) | 0.51(0.60) | 0.53(0.65) | 0.50(0.61) | 0.64(0.60) | 0.31(0.28) | 0.19(0.34) |
| BMMC (Chen et al., 2014) | 0.40(0.51) | 0.44(0.59) | 0.45(0.61) | 0.49(0.66) | 0.66(0.69) | 0.29(0.26) | 0.16(0.32) |
| RIM (Gomes et al., 2010) | 0.50(0.63) | 0.52(0.68) | 0.55(0.71) | 0.51(0.67) | 0.54(0.62) | 0.20(0.25) | 0.17(0.38) |
| RatioRF (Bicego et al., 2021) | 0.54(**0.85**) | 0.55(0.86) | 0.64(0.86) | 0.62(0.86) | 0.48(0.39) | 0.20(0.09) | 0.26(0.38) |
| CDSK (**Ours**) | **0.66(0.85)** | **0.72(0.88)** | **0.68(0.87)** | **0.73(0.89)** | **0.76(0.75)** | **0.46(0.39)** | **0.31(0.41)** |

## 6 CONCLUSION

We propose a new clustering framework termed Clustering by Discriminative Similarity (CDS), which searches for the optimal partition of data where the associated unsupervised classifier has minimum generalization error bound. Under this framework, discriminative similarity is induced by the generalization error bound for unsupervised similarity-based classifier, and CDS minimizes discriminative similarity between different clusters. It is also proved that the discriminative similarity can be induced from kernel density classification. Based on CDS, we propose a new clustering method named CDSK (CDS via unsupervised kernel classification), and demonstrate its effectiveness in data clustering.

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

## A  CONNECTION TO KERNEL DENSITY CLASSIFICATION

In this section, we show that the discriminative similarity (8) can also be induced from kernel density classification with varying weights on the data, and binary classification is considered in this section. For any classification model $M_{\mathcal{Y}} = (\mathcal{S}, f)$ with hypothetical labeling $\mathcal{Y}$ and the labeled data $\mathcal{S} = \{\mathbf{x}_i, y_i\}_{i=1}^n$, suppose the joint distribution $P_{XY}$ over $\mathcal{X} \times \{1, 2\}$ has probabilistic density function $p(\mathbf{x}, y)$. Let $P_X$ be the induced marginal distribution over the data with probabilistic density function $p(\mathbf{x})$. Robust kernel density estimation methods (Girolami & He, 2003; Kim & Scott, 2008; Mahapatruni & Gray, 2011; Kim & Scott, 2012) suggest the following kernel density estimator where the kernel contributions of different data points are reflected by different nonnegative weights that sum up to 1:

$$\widehat{p}(\mathbf{x}) = \tau_0 \sum_{i=1}^n \alpha_i K_\tau(\mathbf{x} - \mathbf{x}_i), \mathbf{1}^\top \boldsymbol{\alpha} = 1, \boldsymbol{\alpha} \geq 0, \tag{13}$$

where $\tau_0 = \frac{1}{(2\pi)^{d/2} h^d}$. Based on (13), it is straightforward to obtain the following kernel density estimator of the density function $p(\mathbf{x}, y)$:

$$\widehat{p}(\mathbf{x}, y) = \tau_0 \sum_{i:y_i=y} \alpha_i K_\tau(\mathbf{x} - \mathbf{x}_i). \tag{14}$$

Kernel density classifier is learnt from the labeled data $\mathcal{S}$ and constructed by kernel density estimators (14). Kernel density classifier resembles the Bayes classifier, and it classifies the test data $\mathbf{x}$ based on the conditional label distribution $P(Y|X = \mathbf{x})$, or equivalently, $\mathbf{x}$ is assigned to class 1 if $\widehat{p}(\mathbf{x}, 1) - \widehat{p}(\mathbf{x}, 2) \geq 0$, otherwise it is assigned to class 2. Intuitively, it is preferred that the decision function $\widehat{r}(\mathbf{x}, \boldsymbol{\alpha}) = \widehat{p}(\mathbf{x}, 1) - \widehat{p}(\mathbf{x}, 2)$ is close to the true Bayes decision function $r = p(\mathbf{x}, 1) - p(\mathbf{x}, 2)$. Girolami & He (2003); Kim & Scott (2008) propose to use Integrated Squared Error (ISE) as the metric to measure the distance between the kernel density estimators and their true counterparts, and the oracle inequality is obtained that relates the performance of the $L_2$ classifier in Kim & Scott (2008) to the best possible performance of kernel density classifier in the same category. ISE is adopted in our analysis of kernel density classification, and the ISE between the decision function $\widehat{r}$ and the true Bayes decision function $r$ is defined as

$$\text{ISE}(\widehat{r}, r) = \|\widehat{r} - r\|_{\mathcal{L}_2}^2 = \int_{\mathbb{R}^d} (\widehat{r} - r)^2 dx. \tag{15}$$

The upper bound for the ISE $\text{ISE}(r, \widehat{r})$ also induces discriminative similarity between the data from different classes, which is presented in the following theorem.

**Theorem A.1.** Let $n_1 = \sum_{i=1} \mathbb{I}_{y_i=1}$ and $n_2 = \sum_{i=1} \mathbb{I}_{y_i=2}$. With probability at least $1 - 2n_2 \exp\big(-2(n-1)\varepsilon^2\big) - 2n \exp\big(-2n\varepsilon^2\big)$ over the labeled data $\mathcal{S}$, the ISE between the decision function $\widehat{r}(\mathbf{x}, \boldsymbol{\alpha})$ and the true Bayes decision function $r(\mathbf{x})$ satisfies

$$\text{ISE}(\widehat{r}, r) \leq \frac{\tau_0}{n} \widehat{\text{ISE}}(\widehat{r}, r) + \tau_1 K(\boldsymbol{\alpha}) + 2\tau_0 \Big(\frac{1}{n-1} + \varepsilon\Big), \tag{16}$$

where

$$\widehat{\text{ISE}}(\widehat{r}, r) = 4 \sum_{1 \leq i < j \leq n} (\alpha_i + \alpha_j) K_\tau(\mathbf{x}_i - \mathbf{x}_j) \mathbb{I}_{y_i \neq y_j} - \sum_{i,j=1}^n (\alpha_i + \alpha_j) K_\tau(\mathbf{x}_i - \mathbf{x}_j), \tag{17}$$

$$K(\boldsymbol{\alpha}) = \boldsymbol{\alpha}^\top (\mathbf{K}_{\sqrt{2}h})\boldsymbol{\alpha} - 4 \sum_{1 \leq i < j \leq n} \alpha_i \alpha_j K_{\sqrt{2}h}(\mathbf{x}_i - \mathbf{x}_j) \mathbb{I}_{y_i \neq y_j}, \tag{18}$$

and $\mathbf{K}_{\sqrt{2}h}$ is the gram matrix evaluated on the data $\{\mathbf{x}_i\}_{i=1}^n$ with the kernel $K_{\sqrt{2}h}$.

Let $\lambda_1 > 0$ be a weighting parameter, then the cost function $\widehat{\mathrm{ISE}} + \lambda_1 K(\boldsymbol{\alpha})$, designed according to the empirical term $\mathrm{ISE}(\widehat{r}, r)$ and the regularization term $K(\boldsymbol{\alpha})$ in the ISE error bound (16), can be expressed as

$$\widehat{\mathrm{ISE}} + \lambda_1 K(\boldsymbol{\alpha}) \leq \sum_{1 \leq i < j \leq n} S_{ij}^{\mathrm{ise}} \mathbb{I}_{y_i \neq y_j} - \sum_{i,j=1}^n (\alpha_i + \alpha_j) K_\tau(\mathbf{x}_i - \mathbf{x}_j) + \lambda_1 \boldsymbol{\alpha}^\top \mathbf{K}_{\sqrt{2}h} \boldsymbol{\alpha},$$

where the first term is comprised of sum of similarity between data from different classes with similarity $S_{ij}^{\mathrm{ise}} = 4(\alpha_i + \alpha_j - \lambda_1 \alpha_i \alpha_j) K_\tau(\mathbf{x}_i - \mathbf{x}_j)$, and $S_{ij}^{\mathrm{ise}}$ is the discriminative similarity induced by the ISE bound for kernel density classification. Note that $S_{ij}^{\mathrm{ise}}$ has the same form as the discriminative similarity $S_{ij}^K$ (8) induced by our CDS model, up to a scaling constant and the choice of the balancing parameter $\lambda$. The proof of Theorem A.1 is deferred to Section E.3.

# B  MORE EXPLANATION ABOUT THE THEORETICAL RESULTS IN SECTION 4.1

## B.1  THEORETICAL SIGNIFICANCE OF THE BOUND (4)

To the best of our knowledge, our generalization error bound (4) is the first principled result about generalization error bound for general similarity-based classifier with strong connection to the established generalization error bound for Support Vector Machines (SVMs) or Kernel Machines.

We now explain this claim. When the similarity function $S$ is a PSD kernel function, we have $S^- \equiv 0, S = S^+$ as explained in Remark 4.3. As a reminder, $S$ is the general similarity function used in the similarity-based classification, and $S^+, S^-$ are PSD kernel functions, and it is proved that $S$ can be decomposed by $S = S^+ - S^-$ under the mild conditions of Lemma 3.1. It follows that we can set $\Omega^-(\boldsymbol{\alpha}) = 0$ and $B^- = 0$. Plugging $B^- = 0$ in the derived generalization error bound for the general similarity-based classification (4), we have

$$\mathrm{Prob}\,[Y \neq f_S(X)] \leq \widehat{R}_n(f_S) + \frac{8R(2c-1)cB^+}{\gamma\sqrt{n}} + \left(\frac{16c(2c-1)B^+ R^2}{\gamma} + 1\right) \sqrt{\frac{\log\frac{4}{\delta}}{2n}}. \tag{19}$$

According to its definition, $\Omega^+(\boldsymbol{\alpha}) = \sum_{y=1}^c \boldsymbol{\alpha}^{(y)^\top} \mathbf{S} \boldsymbol{\alpha}^{(y)}$ because $\mathbf{S} = \mathbf{S}^+$. We define $B := B^+$. Because $\Omega^+(\boldsymbol{\alpha}) \leq B^{+2}$ as mentioned in Theorem 3.2, $B$ satisfies $\sum_{y=1}^c \boldsymbol{\alpha}^{(y)^\top} \mathbf{S} \boldsymbol{\alpha}^{(y)} \leq B^2$. As a result, when $S$ is a PSD kernel function, inequality (19) becomes

$$\mathrm{Prob}\,[Y \neq f_S(X)] \leq \widehat{R}_n(f_S) + \frac{8R(2c-1)cB}{\gamma\sqrt{n}} + \left(\frac{16c(2c-1)R^2 B}{\gamma} + 1\right) \sqrt{\frac{\log\frac{4}{\delta}}{2n}}, \tag{20}$$

with $\sum_{y=1}^c \boldsymbol{\alpha}^{(y)^\top} \mathbf{S} \boldsymbol{\alpha}^{(y)} \leq B^2$.

Note that the bound (20) is in fact the generalization error bound for supervised learning when using $S$ as the similarity function in the similarity-based classification. At the end of this subsection, we provide a lemma proving that $\sum_{y=1}^c \boldsymbol{\alpha}^{(y)^\top} \mathbf{S} \boldsymbol{\alpha}^{(y)} \leq B^2 \Rightarrow \boldsymbol{\alpha}^\top \mathbf{S} \boldsymbol{\alpha} \leq cB^2$, where $c$ is the number of classes.

Now we compare the generalization error bound (20) to the established generalization error bound for Kernel Machines in Bartlett & Mendelson (2003, Theorem 21) for the case that $c = 2$ with

notations adapted to our analysis. The bound in Bartlett & Mendelson (2003, Theorem 21) is for binary classification, which is presented as follows:

$$\text{Prob}\left[Y \neq f_S(X)\right] \leq \widehat{R}_n(f_S) + \frac{4\sqrt{R}B}{\gamma\sqrt{n}} + \left(\frac{8}{\gamma} + 1\right)\sqrt{\frac{\log 4/\delta}{2n}}, \boldsymbol{\alpha}^\top \mathbf{S}\boldsymbol{\alpha} \leq B^2. \quad (21)$$

Comparing our generalization error bound (2) with $c = 2$ to the max-margin generalization error bound (21), it can be easily seen that the two bounds are equivalent up to a constant scaling factor. In fact, our bound (2) is more general which handles multi-class classification.

**Lemma B.1.** When $S$ is a PSD kernel, then $\sum_{y=1}^{c} \boldsymbol{\alpha}^{(y)}^\top \mathbf{S}\boldsymbol{\alpha}^{(y)} \leq B^2 \Rightarrow \boldsymbol{\alpha}^\top \mathbf{S}\boldsymbol{\alpha} \leq cB^2$.

*Proof.* Let $\mathcal{H}$ be the Reproducing Kernel Hilbert Space associated with the PSD kernel function $S$, and $\mathcal{H}$ is also called the feature space associated with $\mathbf{S}$. We use $\langle \cdot, \cdot \rangle_{\mathcal{H}}$ to denote the inner product in the feature space $\mathcal{H}$. Then we have $S(\mathbf{x}_i, \mathbf{x}_j) = \mathbf{S}_{ij} = \langle \Phi(\mathbf{x}_i), \Phi(\mathbf{x}_j) \rangle_{\mathcal{H}}$ where $\Phi$ is the feature mapping associated with $\mathcal{H}$. Because $\boldsymbol{\alpha} = \sum_{y=1}^{c} \boldsymbol{\alpha}^y$, it can be verified that

$$\boldsymbol{\alpha}^\top \mathbf{S}\boldsymbol{\alpha} = \sum_{i=1}^{n}\sum_{j=1}^{n} \alpha_i \alpha_j \mathbf{S}_{ij} = \langle \sum_{i=1}^{n} \alpha_i \Phi(\mathbf{x}_i), \sum_{j=1}^{n} \alpha_j \Phi(\mathbf{x}_j) \rangle \leq c\sum_{y=1}^{c} \langle e_y, e_y \rangle_{\mathcal{H}}, \text{ where } e_y =$$

$\sum_{i:\, y_i=y} \alpha_i^y \Phi(\mathbf{x}_i)$. It follows that $\boldsymbol{\alpha}^\top \mathbf{S}\boldsymbol{\alpha} \leq c\sum_{y=1}^{c} \boldsymbol{\alpha}^{(y)}^\top \mathbf{S}\boldsymbol{\alpha}^{(y)} \leq cB^2$. $\square$

### B.2 Tightness of the Bound

Note that the generalization error $\Pr\left[Y \neq f(X)\right]$ is bounded by $\widehat{R}_n(f) = \frac{1}{n}\sum_{i=1}^{n} \Phi\left(\frac{h_S(\mathbf{x}_i, y_i) - \sum_{y \neq y_i} h_S(\mathbf{x}_i, y)}{\gamma}\right)$ in Theorem 4.2. The underlying principle behind this bound and all such bounds in the statistical machine learning literature such as Bartlett & Mendelson (2003) is the following property about empirical process (adjusted using our notations):

$$\mathfrak{R}(\mathcal{H}) \leq \mathbb{E}_{\mathbf{x},y} \sup_{h \in \mathcal{H}} |\mathbb{E}_{\mathbf{x},y}\widehat{R}_n(f) - R_n(f)| \leq 2\mathfrak{R}(\mathcal{H}), \quad (22)$$

where $\mathbb{E}_{\mathbf{x},y}$ indicates expectation with respect to random couple $(\mathbf{x}, y) \sim P_{XY}$, and $P_{XY}$ is a joint distribution in a discriminative model $M_{\mathcal{Y}} = (\mathcal{S}, f)$. (22) is introduced in the classical properties of empirical process in Gine & Zinn (1984). By Lemma E.2 of this paper, with probability at least $1 - \delta$ over the data $\{\mathbf{x}_i\}_{i=1}^{n} \overset{\text{i.i.d.}}{\sim} P_{XY}$,

$$\mathfrak{R}(\mathcal{H}) \leq \frac{(2c-1)c}{\sqrt{n}}B + \sqrt{2}Bc(2c-1)\sqrt{\frac{\ln\frac{2}{\delta}}{2n}} = \mathcal{O}\left(\frac{B}{\sqrt{n}}\right) \quad (23)$$

for some constant $B$. It follows from (22), (23), and concentration inequality (such as McDiarmid's inequality) that for each sufficiently large $n$, with large probability, $\sup_{h \in \mathcal{H}} |\mathbb{E}_{\mathbf{x},y}\widehat{R}_n(f) - R_n(f)|$ is less than $\mathcal{O}\left(\frac{B}{\sqrt{n}}\right)$. Therefore, we can bound the expectation of the empirical loss, i.e., $\mathbb{E}_{\mathbf{x},y}\widehat{R}_n(f)$, tightly using the empirical loss $R_n(f)$ uniformly over the function space $\mathcal{H}$.

## C Theoretical Properties of the Coordinate Descent Algorithm in Section 5

In this subsection, we give a detailed explanation about the theoretical properties of the coordinate descent algorithm presented in Section 5. We first explain how the objective function of CDSK (10) is connected to the objective function (9) developed in our theoretical analysis. It should be emphasized that (9) cannot be directly used for data clustering since it cannot avoids the trivial solution where all the data are in a single cluster. We adopt the broadly used formulation of normalized

cut and use $\sum_{k=1}^{c} \frac{\mathrm{cut}(\mathbf{A}_k,\bar{\mathbf{A}}_k)}{\mathrm{vol}(\mathbf{A}_k)}$ to replace $\sum_{i<j} S_{ij}^K \mathbb{I}_{y_i \neq y_j}$ in (9), leading to the following optimization problem:

$$\min_{\boldsymbol{\alpha} \in \Lambda, \mathcal{Y}=\{y_i\}_{i=1}^n} \sum_{k=1}^{c} \frac{\mathrm{cut}(\mathbf{A}_k,\bar{\mathbf{A}}_k)}{\mathrm{vol}(\mathbf{A}_k)} - \sum_{i,j=1}^{n} \frac{\alpha_i + \alpha_j}{2} K_\tau(\mathbf{x}_i - \mathbf{x}_j) + \lambda \boldsymbol{\alpha}^\top \mathbf{K} \boldsymbol{\alpha} \triangleq \bar{Q}(\boldsymbol{\alpha}, \mathcal{Y}), \quad (9')$$

where $\{\mathbf{A}\}_{k=1}^c$ are $c$ data clusters according to the cluster labels $\{y_i\}$, $\bar{\mathbf{A}}_k$ is the complement of $\mathbf{A}_k$, $\mathrm{cut}(\mathbf{A}, \mathbf{B}) = \sum_{\mathbf{x}_i \in \mathbf{A}, \mathbf{x}_j \in \mathbf{B}} S_{ij}^K$, $\mathrm{vol}(\mathbf{A}) = \sum_{\mathbf{x}_i \in \mathbf{A}, 1 \leq j \leq n} S_{ij}^K$. We have the following theorem, which can be derived based on Theorem 4.1 in the work of multi-way Cheeger inequalities (Lee et al., 2012).

**Theorem C.1.** $\min_{\boldsymbol{\alpha} \in \Lambda, \mathcal{Y}} \sum_{k=1}^{c} \frac{\mathrm{cut}(\mathbf{A}_k,\bar{\mathbf{A}}_k)}{\mathrm{vol}(\mathbf{A}_k)} \lesssim \sum_{t=1}^{c} \sigma_t(\mathbf{L}^{\mathrm{nor}})$, where $\mathbf{L}^{\mathrm{nor}}$ is the normalized graph Laplacian $\mathbf{L}^{\mathrm{nor}} = (\mathbf{D}^K)^{-1/2} \mathbf{L}^K (\mathbf{D}^K)^{-1/2}$, $a \lesssim b$ indicates $a < Cb$ for some constant $C$ and $\sigma_t(\cdot)$ indicates the $t$-th smallest singular value of a matrix.

Based on Theorem C.1, we resort to solve the following more tractable problem, that is,

$$\min_{\boldsymbol{\alpha} \in \Lambda} \sum_{t=1}^{k} \sigma_t(\mathbf{L}^{\mathrm{nor}}) - \sum_{i,j=1}^{n} \frac{\alpha_i + \alpha_j}{2} K_\tau(\mathbf{x}_i - \mathbf{x}_j) + \lambda \boldsymbol{\alpha}^\top \mathbf{K} \boldsymbol{\alpha} \triangleq Q(\boldsymbol{\alpha}), \quad (9'')$$

because $Q$ (9'') is an upper bound for $\bar{Q}$ (9') up to a constant scaling factor. It can be verified that problem (10) is equivalent to (9''), and (9'') is the underlying optimization problem for data clustering in Section 5. The following proposition shows that the iterative coordinate descent algorithm in Section 5 reduces the value of $Q$ at each iteration.

**Proposition C.2.** The coordinate descent algorithm for problem (10) reduces the value of the objective function $Q(\boldsymbol{\alpha})$ at each iteration.

*Proof.* Let $Q'(\boldsymbol{\alpha}, \mathbf{Y}) \triangleq \mathrm{Tr}(\mathbf{Y}^\top \mathbf{L}^K \mathbf{Y}) - \sum_{i,j=1}^{n} \frac{\alpha_i + \alpha_j}{2} K_\tau(\mathbf{x}_i - \mathbf{x}_j) + \lambda \boldsymbol{\alpha}^\top \mathbf{K} \boldsymbol{\alpha}$, and we use superscript to denote the iteration number of coordinate descent. At iteration $m$, after solving the subproblems (11) and (12), we have $Q'(\boldsymbol{\alpha}^{(m)}, \mathbf{Y}^{(m)}) = Q(\boldsymbol{\alpha}^{(m)})$. At iteration $m+1$, by solving the subproblems (4) and (3) in order again, we have $Q'(\boldsymbol{\alpha}^{(m+1)}, \mathbf{Y}^{(m+1)}) = Q(\boldsymbol{\alpha}^{(m+1)})$. Because of the nature of coordinate descent, $Q'(\boldsymbol{\alpha}^{(m+1)}, \mathbf{Y}^{(m+1)}) \leq Q'(\boldsymbol{\alpha}^{(m)}, \mathbf{Y}^{(m)})$, it follows that $Q(\boldsymbol{\alpha}^{(m+1)}) \leq Q(\boldsymbol{\alpha}^{(m)})$. $\square$

Based on Proposition C.2, the iterations of coordinate descent are similar to that of EM algorithms and they reduce the value of $Q$, where $\mathbf{Y}$ plays the role of latent variable for EM algorithms.

## D    MORE DETAILS ABOUT OPTIMIZATION OF CDSK

The optimization of CDSK comprises $M$ iterations of coordinate descent, wherein each iteration solves the following two subproblems.

1) With constant $\boldsymbol{\alpha}$,

$$\min_{\mathbf{Y} \in \mathbb{R}^{n \times c}} \mathrm{Tr}(\mathbf{Y}^\top \mathbf{L}^K \mathbf{Y}) \quad s.t. \ \mathbf{Y}^\top \mathbf{D}^K \mathbf{Y} = \mathbf{I}_c, \quad (24)$$

2) With constant $\mathbf{Y}$,

$$\min_{\boldsymbol{\alpha} \in \Lambda,} \mathrm{Tr}(\mathbf{Y}^\top \mathbf{L}^K \mathbf{Y}) - \sum_{i,j=1}^{n} \frac{\alpha_i + \alpha_j}{2} K_\tau(\mathbf{x}_i - \mathbf{x}_j) + \lambda \boldsymbol{\alpha}^\top \mathbf{K} \boldsymbol{\alpha}$$
$$s.t. \ \mathbf{Y}^\top \mathbf{D}^K \mathbf{Y} = \mathbf{I}_c, \quad (25)$$

The first subproblem (24) takes $\mathcal{O}(n^2 c)$ steps using truncated Singular Value Decomposition (SVD) by Krylov subspace iterative method. We adopt Sequential Minimal Optimization (SMO) (Platt, 1998) to solve the second subproblem (25), which takes roughly $\mathcal{O}(n^{2.1})$ steps as reported in Platt (1998). SMO is an iterative algorithm where each iteration of SMO solves the quadratic programming (25) with respect to only two elements of the weights $\boldsymbol{\alpha}$, so that each iteration of SMO can be performed efficiently. Therefore, the overall time complexity of CDSK is $\mathcal{O}(Mcn^2 + Mn^{2.1})$.

## E    PROOFS

### E.1    PROOF OF LEMMA 4.1

Before stating the proof of Lemma 4.1, we introduce the famous spectral theorem in operator theory below.

**Theorem E.1. (Spectral Theorem)** Let $L$ be a compact linear operator on a Hilbert space $\mathcal{H}$. Then there exists in H an orthonormal basis $\{\phi_1, \phi_2, \ldots\}$ consisting of eigenvectors of $L$. If $\lambda_k$ is the eigenvalue corresponding to $\phi_k$, then the set $\{\lambda_k\}$ is either finite or $\lambda_k \to 0$ when $k \to \infty$. In addition, the eigenvalues are real if $L$ is self-adjoint.

Recall that the integral operator by $S$ is defined as

$$(L_S f)(\mathbf{x}) = \int S(\mathbf{x}, \mathbf{t}) f(\mathbf{t}) d\mathbf{t},$$

and we are ready to prove Lemma 4.1.

**Proof of Lemma 4.1.** It can be verified that $L_S$ is a compact operator. Therefore, according to Theorem E.1, $\{\phi_k\}$ is an orthogonal basis of $\mathcal{L}^2$. Note that $\phi_k$ is the eigenfunction of $L_S$ with eigenvalue $\lambda_k$ if $L_S \phi_k = \lambda_k \phi_k$.

With fixed $\mathbf{x} \in \mathcal{X}$, we then have

$$\left| \sum_{k=m}^{m+\ell} \lambda_k \phi_k(\mathbf{x}) \phi_k(\mathbf{t}) \right|$$

$$\leq \Big( \sum_{k=m}^{m+\ell} |\lambda_k| |\phi_k(\mathbf{x})|^2 \Big)^{\frac{1}{2}} \cdot \Big( \sum_{k=m}^{m+\ell} |\lambda_k| |\phi_k(\mathbf{t})|^2 \Big)^{\frac{1}{2}}$$

$$\leq \sqrt{C} \Big( \sum_{k=m}^{m+\ell} |\lambda_k| |\phi_k(\mathbf{x})|^2 \Big)^{\frac{1}{2}}.$$

It follows that the series $\sum_{k \geq 1} \lambda_k \phi_k(\mathbf{x}) \phi_k(\mathbf{t})$ converges to a continuous function $e_{\mathbf{x}}$ uniformly on $\mathbf{t}$. This is because $\phi_k = \frac{L_S \phi_k}{\lambda_k}$ is continuous for nonzero $\lambda_k$.

On the other hand, for fixed $\mathbf{x} \in \mathcal{X}$, as a function in $\mathcal{L}^2$,

$$S(\mathbf{x}, \cdot) = \sum_{k \geq 1} \langle S(\mathbf{x}, \cdot), \phi_k \rangle \phi_k = \sum_{k \geq 1} \lambda_k \phi_k(\mathbf{x}) \phi_k(\cdot).$$

Therefore, for fixed $\mathbf{x} \in \mathcal{X}$, $S(\mathbf{x}, \cdot) = \sum_{k \geq 1} \lambda_k \phi_k(\mathbf{x}) \phi_k(\cdot) = e_{\mathbf{x}}(\cdot)$ almost surely w.r.t the Lebesgue measure. Since both are continuous functions, we must have $S(\mathbf{x}, \mathbf{t}) = \sum_{k \geq 1} \lambda_k \phi_k(\mathbf{x}) \phi_k(\mathbf{t})$ for any $\mathbf{t} \in \mathcal{X}$. It follows that $S(\mathbf{x}, \mathbf{t}) = \sum_{k \geq 1} \lambda_k \phi_k(\mathbf{x}) \phi_k(\mathbf{t})$ for any $\mathbf{x}, \mathbf{t} \in \mathcal{X}$.

We now consider two series which correspond to the positive eigenvalues and negative eigenvalues of $L_S$, namely $\sum_{k:\, \lambda_k \geq 0} \lambda_k \phi_k(\mathbf{x}) \phi_k(\cdot)$ and $\sum_{k:\lambda_k < 0} |\lambda_k| \phi_k(\mathbf{x}) \phi_k(\cdot)$. Using similar argument, for fixed

$\mathbf{x}$, both series converge to a continuous function, and we let

$$S^+(\mathbf{x}, \mathbf{t}) = \sum_{k:\,\lambda_k \geq 0} \lambda_k \phi_k(\mathbf{x})\phi_k(\mathbf{t}),$$

$$S^-(\mathbf{x}, \mathbf{t}) = \sum_{k:\lambda_k < 0} |\lambda_k| \phi_k(\mathbf{x})\phi_k(\mathbf{t}).$$

$S^+(\mathbf{x}, \mathbf{t})$ and $S^-(\mathbf{x}, \mathbf{t})$ are continuous function in $\mathbf{x}$ and $\mathbf{t}$. All the eigenvalues of $S^+$ and $S^-$ are nonnegative, and it can be verified that both are PSD kernels since

$$\sum_{i,j=1}^{n} c_i c_j S^+(\mathbf{x}_i, \mathbf{x}_j) = \sum_{i,j=1}^{n} c_i c_j \sum_{k:\,\lambda_k \geq 0} \lambda_k \phi_k(\mathbf{x}_i)\phi_k(\mathbf{x}_j)$$

$$= \sum_{k:\,\lambda_k \geq 0} \lambda_k \sum_{i,j=1}^{n} c_i c_j \phi_k(\mathbf{x}_i)\phi_k(\mathbf{x}_j)$$

$$= \sum_{k:\,\lambda_k \geq 0} \lambda_k (\sum_{i=1}^{n} c_i \phi(\mathbf{x}_i))^2 \geq 0.$$

Similarly argument applies to $S^-$. Therefore, $S$ is decomposed as $S(\mathbf{x}, \mathbf{t}) = S^+(\mathbf{x}, \mathbf{t}) - S^-(\mathbf{x}, \mathbf{t})$.

$\square$

### E.2 PROOF OF THEOREM 4.2

Lemma E.3 will be used in the Proof of Theorem 4.2. The following lemma is introduced for the proof of Lemma E.3, whose proof appears in the end of this subsection.

**Lemma E.2.** The Rademacher complexity of the class $\mathcal{H}_S$ satisfies

$$\mathfrak{R}(\mathcal{H}_S) \leq (2c - 1) \sum_{y=1}^{c} \mathfrak{R}(\mathcal{H}_{S,y}). \tag{26}$$

**Lemma E.3.** Define $\Omega^+(\boldsymbol{\alpha}) = \sum_{y=1}^{c} \boldsymbol{\alpha}^{(y)\top} \mathbf{S}^+ \boldsymbol{\alpha}^{(y)}$ and $\Omega^-(\boldsymbol{\alpha}) = \sum_{y=1}^{c} \boldsymbol{\alpha}^{(y)\top} \mathbf{S}^- \boldsymbol{\alpha}^{(y)}$. When $\Omega^+(\boldsymbol{\alpha}) \leq {B^+}^2, \Omega^-(\boldsymbol{\alpha}) \leq {B^-}^2$ for positive constant $B^+$ and $B^-$, $\sup_{\mathbf{x} \in \mathcal{X}} |S^+(\mathbf{x}, \mathbf{x})| \leq R^2$, $\sup_{\mathbf{x} \in \mathcal{X}} |S^-(\mathbf{x}, \mathbf{x})| \leq R^2$ for some $R > 0$, then with probability at least $1 - \delta$ over the data $\{\mathbf{x}_i\}_{i=1}^{n}$, the Rademacher complexity of the class $\mathcal{H}_S$ satisfies

$$\mathfrak{R}(\mathcal{H}_S) \leq \frac{R(2c-1)c(B^+ + B^-)}{\sqrt{n}} + 2c(2c-1)(B^+ + B^-)R^2 \sqrt{\frac{\ln \frac{2}{\delta}}{2n}}. \tag{27}$$

**Proof of Lemma E.3**. According to Lemma 4.1, $S$ is decomposed into two PSD kernels as $S = S^+ - S^-$. Therefore, the are two Reproducing Kernel Hilbert Spaces $\mathcal{H}_S^+$ and $\mathcal{H}_S^-$ that are associated with $S^+$ and $S^-$ respectively, and the canonical feature mappings in $\mathcal{H}_S^+$ and $\mathcal{H}_S^-$ are $\phi^+$ and $\phi^-$, with $S^+(\mathbf{x}, \mathbf{t}) = \langle \phi^+(\mathbf{x}), \phi^+(\mathbf{t}) \rangle_{H_K^+}$ and $S^-(\mathbf{x}, \mathbf{t}) = \langle \phi^-(\mathbf{x}), \phi^-(\mathbf{t}) \rangle_{H_K^-}$. In the following text, we will omit the subscripts $H_K^+$ and $H_K^-$ without confusion.

For any $1 \leq y \leq c$,

$$h_S(\mathbf{x}, y) = \sum_{i:\,y_i = y} \alpha_i S(\mathbf{x}, \mathbf{x}_i) = \langle \mathbf{w}^+, \phi^+(\mathbf{x}) \rangle - \langle \mathbf{w}^-, \phi^-(\mathbf{x}) \rangle$$

with $\|\mathbf{w}^+\|^2 = \boldsymbol{\alpha}^{(y)\top} \mathbf{S}^+ \boldsymbol{\alpha}^{(y)} \leq {B^+}^2$ and $\|\mathbf{w}^-\|^2 = \boldsymbol{\alpha}^{(y)\top} \mathbf{S}^- \boldsymbol{\alpha}^{(y)} \leq {B^-}^2$. Therefore,

$$\mathcal{H}_{S,y} \subseteq \tilde{\mathcal{H}}_{S,y} = \{(\mathbf{x}, y) \to \langle \mathbf{w}^+, \phi^+(\mathbf{x}) \rangle - \langle \mathbf{w}^-, \phi^-(\mathbf{x}) \rangle,$$

$$\|\mathbf{w}^+\|^2 \leq {B^+}^2, \|\mathbf{w}^-\|^2 \leq {B^-}^2\}, 1 \leq y \leq c,$$

and $\mathfrak{R}(\mathcal{H}_{S,y}) \subseteq \mathfrak{R}(\tilde{\mathcal{H}}_{S,y})$. Since we are deriving upper bound for $\mathfrak{R}(\mathcal{H}_{S,y})$, we slightly abuse the notation and let $\mathcal{H}_{S,y}$ represent $\tilde{\mathcal{H}}_{S,y}$ in the remaining part of this proof.

For $\mathbf{x}, \mathbf{t} \in \mathbb{R}^d$ and any $h_S \in \mathcal{H}_{S,y}$, we have

$$
\begin{aligned}
|h_S(\mathbf{x}) - h_S(\mathbf{t})| &= |\langle \mathbf{w}^+, \phi^+(\mathbf{x})\rangle - \langle \mathbf{w}^-, \phi^-(\mathbf{x})\rangle - \langle \mathbf{w}^+, \phi^+(\mathbf{t})\rangle + \langle \mathbf{w}^-, \phi^-(\mathbf{t})\rangle| \\
&= |\langle \mathbf{w}^+, \phi^+(\mathbf{x}) - \phi^+(\mathbf{t})\rangle + \langle \mathbf{w}^-, \phi^-(\mathbf{t}) - \phi^-(\mathbf{x})\rangle| \\
&\leq B^+ \|\phi^+(\mathbf{x}) - \phi^+(\mathbf{t})\| + B^- \|\phi^-(\mathbf{x}) - \phi^-(\mathbf{t})\|) \\
&\leq (B^+ + B^-)\sqrt{S^+(\mathbf{x},\mathbf{x}) + S^+(\mathbf{t},\mathbf{t}) + 2\sqrt{S^+(\mathbf{x},\mathbf{x})S^+(\mathbf{t},\mathbf{t})}} \\
&\leq 2R^2(B^+ + B^-).
\end{aligned}
$$

We now approximate the Rademacher complexity of the function class $\mathcal{H}_{S,y}$ with its empirical version $\widehat{\mathfrak{R}}(\mathcal{H}_{S,y})$ using the sample $\{\mathbf{x}_i\}$. For each $1 \leq y \leq c$, Define $E_{\{\mathbf{x}_i\}}^{(y)} = \widehat{\mathfrak{R}}(\mathcal{H}_{S,y}) = \mathbb{E}_{\{\sigma_i\}}\left[\sup_{h_S(\cdot,y)\in\mathcal{H}_{S,y}} \left|\frac{1}{n}\sum_{i=1}^{n} \sigma_i h_S(\mathbf{x}_i, y)\right|\right]$, then $\sum_{y=1}^{c} \mathfrak{R}(\mathcal{H}_{S,y}) = \mathbb{E}_{\{\mathbf{x}_i\}}\left[\sum_{y=1}^{c} E_{\{\mathbf{x}_i\}}^{(y)}\right]$, and

$$
\begin{aligned}
&\sup_{\mathbf{x}_1,\ldots,\mathbf{x}_n,\mathbf{x}_t'} \left|E_{\mathbf{x}_1,\ldots,\mathbf{x}_{t-1},\mathbf{x}_t,\mathbf{x}_{t+1},\ldots,\mathbf{x}_n}^{(y)} - E_{\mathbf{x}_1,\ldots,\mathbf{x}_{t-1},\mathbf{x}_t',\mathbf{x}_{t+1},\ldots,\mathbf{x}_n}^{(y)}\right| \\
&= \sup_{\mathbf{x}_1,\ldots,\mathbf{x}_n,\mathbf{x}_t'} \left|\mathbb{E}_{\{\sigma_i\}}\left[\sup_{h_S(\cdot,y)\in\mathcal{H}_{S,y}}\left|\frac{1}{n}\sum_{i=1}^{n}\sigma_i h_S(\mathbf{x}_i,y)\right| - \sup_{h_S(\cdot,y)\in\mathcal{H}_{S,y}}\left|\frac{1}{n}\sum_{i\neq t}\sigma_i h_S(\mathbf{x}_i,y) + \frac{h_S(\mathbf{x}_t',y)}{n}\right|\right]\right| \\
&\leq \sup_{\mathbf{x}_1,\ldots,\mathbf{x}_n,\mathbf{x}_t'} \mathbb{E}_{\{\sigma_i\}}\left[\left|\sup_{h_S(\cdot,y)\in\mathcal{H}_{S,y}}\left|\frac{1}{n}\sum_{i=1}^{n}\sigma_i h_S(\mathbf{x}_i,y)\right| - \sup_{h_S(\cdot,y)\in\mathcal{H}_{S,y}}\left|\frac{1}{n}\sum_{i\neq t}\sigma_i h_S(\mathbf{x}_i,y) + \frac{h_S(\mathbf{x}_t',y)}{n}\right|\right|\right] \\
&\leq \sup_{\mathbf{x}_1,\ldots,\mathbf{x}_n,\mathbf{x}_t'} \mathbb{E}_{\{\sigma_i\}}\left[\sup_{h_S(\cdot,y)\in\mathcal{H}_{S,y}}\left|\left|\frac{1}{n}\sum_{i=1}^{n}\sigma_i h_S(\mathbf{x}_i,y)\right| - \left|\frac{1}{n}\sum_{i\neq t}\sigma_i h_S(\mathbf{x}_i,y) + \frac{h_S(\mathbf{x}_t',y)}{n}\right|\right|\right] \\
&\leq \sup_{\mathbf{x}_1,\ldots,\mathbf{x}_n,\mathbf{x}_t'} \mathbb{E}_{\{\sigma_i\}}\left[\sup_{h_S(\cdot,y)\in\mathcal{H}_{S,y}}\left|\frac{1}{n}\sum_{i=1}^{n}\sigma_i h_S(\mathbf{x}_i,y) - \left(\frac{1}{n}\sum_{i\neq t}\sigma_i h_S(\mathbf{x}_i,y) + \frac{h_S(\mathbf{x}_t',y)}{n}\right)\right|\right] \\
&= \sup_{\mathbf{x}_t,\mathbf{x}_t'} \mathbb{E}_{\{\sigma_i\}}\left[\sup_{h_S(\cdot,y)\in\mathcal{H}_{S,y}}\left|\frac{h_S(\mathbf{x}_t,y)}{n} - \frac{h_S(\mathbf{x}_t',y)}{n}\right|\right] \\
&\leq \frac{2R^2(B^+ + B^-)}{n}.
\end{aligned}
$$

It follows that $\left|\sum_{y=1}^{c} E_{\mathbf{x}_1,\ldots,\mathbf{x}_{t-1},\mathbf{x}_t,\mathbf{x}_{t+1},\ldots,\mathbf{x}_n}^{(y)} - \sum_{y=1}^{c} E_{\mathbf{x}_1,\ldots,\mathbf{x}_{t-1},\mathbf{x}_t',\mathbf{x}_{t+1},\ldots,\mathbf{x}_n}^{(y)}\right| \leq \frac{2R^2(B^+ + B^-)c}{n}$. According to the McDiarmid's Inequality,

$$
\Pr\left[\left|\sum_{y=1}^{c}\widehat{\mathfrak{R}}(\mathcal{H}_{S,y}) - \sum_{y=1}^{c}\mathfrak{R}(\mathcal{H}_{S,y})\right| \geq \varepsilon\right] \leq 2\exp\left(-\frac{n\varepsilon^2}{2(B^+ + B^-)^2 R^4 c^2}\right). \tag{28}
$$

Now we derive the upper bound for the empirical Rademacher complexity:

$$\sum_{y=1}^{c} \widehat{\mathfrak{R}}(\mathcal{H}_{S,y}) = \sum_{y=1}^{c} \mathbb{E}_{\{\sigma_i\}} \left[ \sup_{h_S \in \mathcal{H}_{S,y}} \left| \frac{1}{n} \sum_{i=1}^{n} \sum_{j=1}^{n} \sigma_i h_S(\mathbf{x}_i) \right| \right] \tag{29}$$

$$\leq \frac{1}{n} \sum_{y=1}^{c} \mathbb{E}_{\{\sigma_i\}} \left[ \sup_{\|\mathbf{w}^+\| \leq B^+, \|\mathbf{w}^-\| \leq B^-} \left| \sum_{i=1}^{n} \sigma_i \left( \langle \mathbf{w}^+, \phi^+(\mathbf{x}_i) \rangle - \langle \mathbf{w}^-, \phi^-(\mathbf{x}_i) \rangle \right) \right| \right]$$

$$\leq \frac{1}{n} \sum_{y=1}^{c} \mathbb{E}_{\{\sigma_i\}} \left[ B^+ \| \sum_{i=1}^{n} \sigma_i \phi^+(\mathbf{x}_i) \| + B^- \| \sum_{i=1}^{n} \sigma_i \phi^-(\mathbf{x}_i) \| \right]$$

$$= \frac{B^+ c}{n} \mathbb{E}_{\{\sigma_i\}} \left[ \| \sum_{i=1}^{n} \sigma_i \phi^+(\mathbf{x}_i) \| \right] + \frac{B^- c}{n} \mathbb{E}_{\{\sigma_i\}} \left[ \| \sum_{i=1}^{n} \sigma_i \phi^-(\mathbf{x}_i) \| \right]$$

$$\leq \frac{B^+ c}{n} \sqrt{\mathbb{E}_{\{\sigma_i\}} \left[ \| \sum_{i=1}^{n} \sigma_i \phi^+(\mathbf{x}_i) \|^2 \right]} + \frac{B^- c}{n} \sqrt{\mathbb{E}_{\{\sigma_i\}} \left[ \| \sum_{i=1}^{n} \sigma_i \phi^-(\mathbf{x}_i) \|^2 \right]}$$

$$\leq \frac{B^+ c}{n} \sqrt{\sum_{i}^{n} s^+(\mathbf{x}_i, \mathbf{x}_i)} + \frac{B^- c}{n} \sqrt{\sum_{i}^{n} s^-(\mathbf{x}_i, \mathbf{x}_i)} \leq \frac{Rc}{\sqrt{n}} (B^+ + B^-).$$

By Lemma E.2, (28) and (29), with probability at least $1 - \delta$, we have

$$\mathfrak{R}(\mathcal{H}_S) \leq (2c-1) \sum_{y=1}^{c} \mathfrak{R}(\mathcal{H}_{S,y}) \leq \frac{R(2c-1)c(B^+ + B^-)}{\sqrt{n}} + 2c(2c-1)(B^+ + B^-)R^2 \sqrt{\frac{\ln \frac{2}{\delta}}{2n}}. \tag{30}$$

$\square$

**Proof of Theorem 4.2** . According to Theorem 2 in Koltchinskii & Panchenko (2002), with probability $1 - \delta$ over the labeled data $\mathcal{S}$ with respect to any distribution in $\mathcal{P}$, the generalization error of the kernel classifier $f_S$ satisfies

$$R(f_S) \leq \widehat{R}_n(f_S) + \frac{8}{\gamma} \mathfrak{R}(\mathcal{H}_S) + \sqrt{\frac{\ln 2/\delta}{2n}}, \tag{31}$$

where $\widehat{R}_n(f_S) = \frac{1}{n} \sum_{i=1}^{n} \Phi(\frac{m_{h_S}(\mathbf{x}_i, y_i)}{\gamma})$ is empirical error of the classifier for $\gamma > 0$. Due to the facts that $m_{h_S}(\mathbf{x}, y) = h_S(\mathbf{x}_i, y_i) - \text{argmax}_{y' \neq y_i} h_S(\mathbf{x}_i, y')$, $\boldsymbol{\alpha}$ is a positive vector and $S$ is nonnegative, we have $m_{h_S}(\mathbf{x}, y) \geq h_S(\mathbf{x}_i, y_i) - \sum_{y \neq y_i} h_S(\mathbf{x}_i, y)$. Note that $\Phi(\cdot)$ is a non-increasing function, it follows that

$$\Phi\left(\frac{m_{h_S}(\mathbf{x}_i, y_i)}{\gamma}\right) \leq \Phi\left(\frac{h_S(\mathbf{x}_i, y_i) - \sum\limits_{y \neq y_i} h_S(\mathbf{x}_i, y)}{\gamma}\right). \tag{32}$$

Applying Lemma E.3, (4) holds with probability $1 - \delta$. When $\gamma \geq 1$, it can be verified that $\left| h_S(\mathbf{x}_i, y_i) - \sum\limits_{y \neq y_i} h_S(\mathbf{x}_i, y) \right| \leq \sum\limits_{y=1}^{c} h_S(\mathbf{x}_i, y) \leq \sum\limits_{i=1}^{n} \alpha_i = 1 \leq \gamma$ for all $(\mathbf{x}_i, y_i)$, so that

$$\Phi\left(\frac{h_S(\mathbf{x}_i, y_i) - \sum\limits_{y \neq y_i} h_S(\mathbf{x}_i, y)}{\gamma}\right) \leq 1 - \frac{h(\mathbf{x}_i, y_i) - \sum\limits_{y \neq y_i} h(\mathbf{x}_i, y)}{\gamma}. \tag{33}$$

Note that when $h_S(\mathbf{x}_i, y_i) - \sum_{y \neq y_i} h_S(\mathbf{x}_i, y) \geq 0$, then $\frac{1}{\gamma}\left[h_S(\mathbf{x}_i, y_i) - \sum_{y \neq y_i} h_S(\mathbf{x}_i, y)\right] \in [0, 1]$ so

that $\Phi\left(\frac{h_S(\mathbf{x}_i, y_i) - \sum_{y \neq y_i} h_S(\mathbf{x}_i, y)}{\gamma}\right) = 1 - \frac{h(\mathbf{x}_i, y_i) - \sum_{y \neq y_i} h(\mathbf{x}_i, y)}{\gamma}$. If $h_S(\mathbf{x}_i, y_i) - \sum_{y \neq y_i} h_S(\mathbf{x}_i, y) < 0$, we

have $\Phi\left(\frac{h_S(\mathbf{x}_i, y_i) - \sum_{y \neq y_i} h_S(\mathbf{x}_i, y)}{\gamma}\right) \leq 1 \leq 1 - \frac{h(\mathbf{x}_i, y_i) - \sum_{y \neq y_i} h(\mathbf{x}_i, y)}{\gamma}$. Therefore, (33) always holds.

By the definition of $\widehat{R}_n(f_S)$, (32), and (33), (5) is obtained.

$\square$

**Remark E.4.** It can be verified that the image of the similarity function $S$ in Lemma 4.1 and Theorem 4.2 can be generalized from $[0, 1]$ to $[0, a]$ for any $a \in \mathbb{R}, a > 0$ with the condition $\gamma \geq 1$ replaced by $\gamma \geq a$. This is because $L_S$ is a compact operator for continuous similarity function $S \colon \mathcal{X} \times \mathcal{X} \to [0, a]$, and $\left|h_S(\mathbf{x}_i, y_i) - \sum_{y \neq y_i} h_S(\mathbf{x}_i, y)\right| \leq \sum_{y=1}^{c} h_S(\mathbf{x}_i, y) \leq a$. Furthermore, given a symmetric and continuous function $S \colon \mathcal{X} \times \mathcal{X} \to [c, d], c, d \in \mathbb{R}, c < d$, we can obtain a symmetric and continuous function $S' \colon \mathcal{X} \times \mathcal{X} \to [0, 1]$ by setting $S' = \frac{S-a}{b-a}$, and then apply all the theoretical results of this paper to CDS with $S'$ being the similarity function for the similarity-based classifier.

**Proof of Lemma E.2.** Inspired by Koltchinskii & Panchenko (2002), we first prove that the Rademacher complexity of the function class formed by the maximum of several hypotheses is bounded by two times the sum of the Rademacher complexity of the function classes that these hypothesis belong to. That is,

$$\mathfrak{R}(\mathcal{H}_{\max}) \leq 2 \sum_{y=1}^{k} \mathfrak{R}(\mathcal{H}_{S,y}), \tag{34}$$

where $\mathcal{H}_{\max} = \{\max\{h_1, \ldots, h_k\} \colon h_y \in \mathcal{H}_{S,y}, 1 \leq y \leq k\}$ for $1 \leq k \leq c - 1$.

If no confusion arises, the notations $(\{\sigma_i\}, \{\mathbf{x}_i, y_i\})$ are omitted in the subscript of the expectation operator in the following text, i.e., $\mathbb{E}_{\{\sigma_i\}, \{\mathbf{x}_i, y_i\}}$ is abbreviated to $\mathbb{E}$. According to Theorem 11 of Koltchinskii & Panchenko (2002), it can be verified that

$$\mathbb{E}_{\{\sigma_i\}, \{\mathbf{x}_i, y_i\}}\left[\left(\sup_{h \in \mathcal{H}_{\max}}\left|\frac{1}{n}\sum_{i=1}^{n}\sigma_i h_S(\mathbf{x}_i)\right|\right)^+\right]$$
$$\leq \sum_{y=1}^{k}\mathbb{E}_{\{\sigma_i\}, \{\mathbf{x}_i, y_i\}}\left[\left(\sup_{h \in \mathcal{H}_{S,y}}\left|\frac{1}{n}\sum_{i=1}^{n}\sigma_i h_S(\mathbf{x}_i)\right|\right)^+\right].$$

Therefore,

$$\mathfrak{R}(\mathcal{H}_{\max}) = \mathbb{E}_{\{\sigma_i\}, \{\mathbf{x}_i, y_i\}}\left[\sup_{h \in \mathcal{H}_{\max}}\left|\frac{1}{n}\sum_{i=1}^{n}\sigma_i h_S(\mathbf{x}_i)\right|\right]$$
$$\leq \mathbb{E}_{\{\sigma_i\}, \{\mathbf{x}_i, y_i\}}\left[\left(\sup_{h \in \mathcal{H}_{\max}}\frac{1}{n}\sum_{i=1}^{n}\sigma_i h_S(\mathbf{x}_i)\right)^+\right]$$
$$+ \mathbb{E}_{\{\sigma_i\}, \{\mathbf{x}_i, y_i\}}\left[\left(\sup_{h \in \mathcal{H}_{\max}}-\frac{1}{n}\sum_{i=1}^{n}\sigma_i h_S(\mathbf{x}_i)\right)^+\right]$$
$$= 2\mathbb{E}_{\{\sigma_i\}, \{\mathbf{x}_i, y_i\}}\left[\left(\sup_{h \in \mathcal{H}_{\max}}\frac{1}{n}\sum_{i=1}^{n}\sigma_i h_S(\mathbf{x}_i)\right)^+\right]$$
$$\leq 2\sum_{y=1}^{k}\mathbb{E}_{\{\sigma_i\}, \{\mathbf{x}_i, y_i\}}\left[\left(\sup_{h \in \mathcal{H}_{S,y}}\frac{1}{n}\sum_{i=1}^{n}\sigma_i h_S(\mathbf{x}_i)\right)^+\right]$$

$$\leq 2 \sum_{y=1}^{k} \mathbb{E}_{\{\sigma_i\},\{\mathbf{x}_i,y_i\}} \left[ \sup_{h \in \mathcal{H}_{S,y}} \left| \frac{1}{n} \sum_{i=1}^{n} \sigma_i h_S(\mathbf{x}_i) \right| \right] = 2 \sum_{y=1}^{k} \mathfrak{R}(\mathcal{H}_{S,y}). \qquad (35)$$

The equality in the third line of (35) is due to the fact that $-\sigma_i$ has the same distribution as $\sigma_i$. Using this fact again, (34), we have

$$\mathfrak{R}(\mathcal{H}_S) = \mathbb{E}_{\{\sigma_i\},\{\mathbf{x}_i,y_i\}} \left[ \sup_{m_{h_S} \in \mathcal{H}_S} \left| \frac{1}{n} \sum_{i=1}^{n} \sigma_i m_{h_S}(\mathbf{x}_i, y_i) \right| \right]$$

$$= \mathbb{E}_{\{\sigma_i\},\{\mathbf{x}_i,y_i\}} \left[ \sup_{m_{h_S} \in \mathcal{H}_S} \left| \frac{1}{n} \sum_{i=1}^{n} \sigma_i \sum_{y=1}^{c} m_{h_S}(\mathbf{x}_i, y) \mathbb{1}_{y=y_i} \right| \right]$$

$$\leq \sum_{y=1}^{c} \mathbb{E}_{\{\sigma_i\},\{\mathbf{x}_i,y_i\}} \left[ \sup_{m_{h_S} \in \mathcal{H}_S} \left| \frac{1}{n} \sum_{i=1}^{n} \sigma_i m_{h_S}(\mathbf{x}_i, y) \mathbb{1}_{y=y_i} \right| \right]$$

$$\leq \frac{1}{2n} \sum_{y=1}^{c} \mathbb{E}_{\{\sigma_i\},\{\mathbf{x}_i,y_i\}} \left[ \sup_{m_{h_S} \in \mathcal{H}_S} \left| \sum_{i=1}^{n} \sigma_i m_{h_S}(\mathbf{x}_i, y)(2\mathbb{1}_{y=y_i} - 1) \right| \right]$$

$$+ \frac{1}{2n} \sum_{y=1}^{c} \mathbb{E}_{\{\sigma_i\},\{\mathbf{x}_i\}} \left[ \sup_{m_{h_S} \in \mathcal{H}_S} \left| \sum_{i=1}^{n} \sigma_i m_{h_S}(\mathbf{x}_i, y) \right| \right]$$

$$= \frac{1}{n} \sum_{y=1}^{c} \mathbb{E}_{\{\sigma_i\},\{\mathbf{x}_i\}} \left[ \sup_{m_{h_S} \in \mathcal{H}_S} \left| \sum_{i=1}^{n} \sigma_i m_{h_S}(\mathbf{x}_i, y) \right| \right]. \qquad (36)$$

Also, for any given $1 \leq y \leq c$,

$$\frac{1}{n} \mathbb{E}_{\{\sigma_i\},\{\mathbf{x}_i\}} \left[ \sup_{m_{h_S} \in \mathcal{H}_S} \left| \sum_{i=1}^{n} \sigma_i m_{h_S}(\mathbf{x}_i, y) \right| \right]$$

$$= \frac{1}{n} \mathbb{E}_{\{\sigma_i\},\{\mathbf{x}_i\}} \left[ \sup_{h_S(\cdot,y) \in \mathcal{H}_{S,y}, y=1...c} \left| \sum_{i=1}^{n} \sigma_i h_S(\mathbf{x}_i, y) - \sigma_i \mathrm{argmax}_{y' \neq y} h_S(\mathbf{x}_i, y') \right| \right]$$

$$\leq \frac{1}{n} \mathbb{E}_{\{\sigma_i\},\{\mathbf{x}_i\}} \left[ \sup_{h_S(\cdot,y) \in \mathcal{H}_{S,y}} \left| \sum_{i=1}^{n} \sigma_i h_S(\mathbf{x}_i, y) \right| \right]$$

$$+ \frac{1}{n} \mathbb{E}_{\{\sigma_i\},\{\mathbf{x}_i\}} \left[ \sup_{h_S(\cdot,y') \in \mathcal{H}'_{S,y}, y' \neq y} \left| \sum_{i=1}^{n} \sigma_i \mathrm{argmax}_{y' \neq y} h_S(\mathbf{x}_i, y') \right| \right]$$

$$\leq \frac{1}{n} \mathbb{E}_{\{\sigma_i\},\{\mathbf{x}_i\}} \left[ \sup_{h_S(\cdot,y) \in \mathcal{H}_{S,y}} \left| \sum_{i=1}^{n} \sigma_i h_S(\mathbf{x}_i, y) \right| \right] + \frac{2}{n} \sum_{y' \neq y} \mathbb{E}_{\{\sigma_i\},\{\mathbf{x}_i\}} \left[ \sup_{h_S(\cdot,y') \in \mathcal{H}'_{S,y}} \left| \sum_{i=1}^{n} \sigma_i h_S(\mathbf{x}_i, y') \right| \right]. \qquad (37)$$

Combining (36) and (37),

$$\mathfrak{R}(\mathcal{H}_S) \leq \sum_{y=1}^{c} \frac{1}{n} \mathbb{E}_{\{\sigma_i\},\{\mathbf{x}_i\}} \left[ \sup_{h_S(\cdot,y) \in \mathcal{H}_{S,y}} \left| \sum_{i=1}^{n} \sigma_i h_S(\mathbf{x}_i, y) \right| \right]$$

$$+ \sum_{y=1}^{c} \sum_{y=1}^{c} \frac{2}{n} \sum_{y' \neq y} \mathbb{E}_{\{\sigma_i\},\{\mathbf{x}_i\}} \left[ \sup_{h_S(\cdot,y') \in \mathcal{H}'_{S,y}} \left| \sum_{i=1}^{n} \sigma_i h_S(\mathbf{x}_i, y') \right| \right]$$

$$= (2c - 1) \sum_{y=1}^{c} \mathbb{E}_{\{\sigma_i\},\{\mathbf{x}_i\}} \left[ \sup_{h_S(\cdot,y) \in \mathcal{H}_{S,y}} \left| \frac{1}{n} \sum_{i=1}^{n} \sigma_i h_S(\mathbf{x}_i, y) \right| \right]$$

$$= (2c - 1) \sum_{y=1}^{c} \mathfrak{R}(\mathcal{H}_{S,y}). \qquad (38)$$

$\square$

### E.3 PROOF OF THEOREM A.1

*Proof.* According to definition of ISE,

$$\text{ISE}(\widehat{r}, r) = \int_{\mathbb{R}^d} (\widehat{r} - r)^2 dx = \int_{\mathbb{R}^d} \widehat{r}(\mathbf{x}, \boldsymbol{\alpha})^2 dx - 2 \int_{\mathbb{R}^d} \widehat{r}(\mathbf{x}, \boldsymbol{\alpha}) r(\mathbf{x}) dx + \int_{\mathbb{R}^d} r(\mathbf{x})^2 dx. \quad (39)$$

For a given distribution, $\int_{\mathbb{R}^d} r(\mathbf{x})^2 dx$ is a constant. By Gaussian convolution theorem,

$$\int_{\mathbb{R}^d} \widehat{r}(\mathbf{x}, \boldsymbol{\alpha})^2 dx = \tau_1 \sum_{y=1}^{2} \boldsymbol{\alpha}^{(y)\top} (\mathbf{K}_{\sqrt{2}h}) \boldsymbol{\alpha}^{(y)} - \tau_1 \sum_{1 \leq i < j \leq n} 2\alpha_i \alpha_j \mathbf{K}_{\sqrt{2}h}(\mathbf{x}_i - \mathbf{x}_j) \mathbb{I}_{y_i \neq y_j}, \quad (40)$$

where $\tau_1 = \frac{1}{(2\pi)^{d/2}(\sqrt{2}h)^d}$. Moreover,

$$\int_{\mathbb{R}^d} \widehat{r}(\mathbf{x}, \boldsymbol{\alpha}) r(\mathbf{x}) dx$$
$$= \int_{\mathbb{R}^d} \widehat{p}(\mathbf{x}, 1) p(\mathbf{x}, 1) dx + \int_{\mathbb{R}^d} \widehat{p}(\mathbf{x}, 2) p(\mathbf{x}, 2) dx - \int_{\mathbb{R}^d} \widehat{p}(\mathbf{x}, 1) p(\mathbf{x}, 2) dx - \int_{\mathbb{R}^d} \widehat{p}(\mathbf{x}, 2) p(\mathbf{x}, 1) dx. \quad (41)$$

Note that

$$\frac{1}{\tau_0} \int_{\mathbb{R}^d} \widehat{p}(\mathbf{x}, 1) p(\mathbf{x}, 1) dx = \sum_{j: \, y_j = 1} \int_{\mathbb{R}^d} \alpha_j K_\tau(\mathbf{x} - \mathbf{x}_j) p(\mathbf{x}, 1) dx,$$

we then use the empirical term $\frac{\sum_{i: \, i \neq j} \alpha_j K_\tau(\mathbf{x}_i - \mathbf{x}_j) \mathbb{I}_{y_i=1}}{n-1}$ to approximate the integral $\int_{\mathbb{R}^d} \alpha_j K_\tau(\mathbf{x} - \mathbf{x}_j) p(\mathbf{x}, 1) dx$. Since $\mathbb{E}_{\{\mathbf{x}_i, y_i\}_{i \neq j}} \left[ \frac{\sum_{i: \, i \neq j} \alpha_j K_\tau(\mathbf{x}_i - \mathbf{x}_j) \mathbb{I}_{y_i=1}}{n-1} \right] = \int_{\mathbb{R}^d} \alpha_j K_\tau(\mathbf{x} - \mathbf{x}_j) p(\mathbf{x}, 1) dx$, and bounded difference holds for $\frac{\sum_{i: \, i \neq j} \alpha_j K_\tau(\mathbf{x}_i - \mathbf{x}_j) \mathbb{I}_{y_i=1}}{n-1}$, therefore,

$$\Pr \left[ \left| \frac{\sum_{i: \, i \neq j} \alpha_j K_\tau(\mathbf{x}_i - \mathbf{x}_j) \mathbb{I}_{y_i=1}}{n-1} - \int_{\mathbb{R}^d} \alpha_j K_\tau(\mathbf{x} - \mathbf{x}_j) p(\mathbf{x}, 1) dx \right| \geq \alpha_j \varepsilon \right] \leq 2 \exp \left( -2(n-1)\varepsilon^2 \right).$$

It follows that with probability at least $1 - 2n_1 \exp \left( -2(n-1)\varepsilon^2 \right)$, where $n_i$ is the number of data points with label $i$,

$$\left| \frac{\sum_{i,j: \, i \neq j, y_i = y_j = 1} \alpha_j K_\tau(\mathbf{x}_i - \mathbf{x}_j)}{n-1} - \frac{1}{\tau_0} \int_{\mathbb{R}^d} \widehat{p}(\mathbf{x}, 1) p(\mathbf{x}, 1) dx \right| \leq \sum_{j: \, y_j = 1} \alpha_j \varepsilon. \quad (42)$$

Similarly, with probability at least $1 - 2n_2 \exp \left( -2(n-1)\varepsilon^2 \right)$,

$$\left| \frac{\sum_{i,j: \, i \neq j, y_i = y_j = 2} \alpha_j K_\tau(\mathbf{x}_i - \mathbf{x}_j)}{n-1} - \frac{1}{\tau_0} \int_{\mathbb{R}^d} \widehat{p}(\mathbf{x}, 2) p(\mathbf{x}, 2) dx \right| \leq \sum_{j: \, y_j = 2} \alpha_j \varepsilon. \quad (43)$$

It follows from (42) and (43) that with probability at least $1 - 2n \exp \left( -2(n-1)\varepsilon^2 \right)$,

$$\left| \frac{\sum_{i,j: \, i \neq j, y_i = y_j} \alpha_j K_\tau(\mathbf{x}_i - \mathbf{x}_j)}{n-1} - \frac{1}{\tau_0} \int_{\mathbb{R}^d} \left( \widehat{p}(\mathbf{x}, 1) p(\mathbf{x}, 1) + \widehat{p}(\mathbf{x}, 2) p(\mathbf{x}, 2) \right) dx \right| \leq \varepsilon. \quad (44)$$

In the same way, with probability at least $1 - 2n \exp \left( -2n\varepsilon^2 \right)$,

$$\left| \frac{\sum_{i,j: \, y_i \neq y_j} \alpha_j K_\tau(\mathbf{x}_i - \mathbf{x}_j)}{n} - \frac{1}{\tau_0} \int_{\mathbb{R}^d} \left( \widehat{p}(\mathbf{x}, 1) p(\mathbf{x}, 2) + \widehat{p}(\mathbf{x}, 2) p(\mathbf{x}, 1) \right) dx \right| \leq \varepsilon. \quad (45)$$

Based on (44) and (45), with probability at least $1 - 2n_2 \exp\left(-2(n-1)\varepsilon^2\right) - 2n \exp\left(-2n\varepsilon^2\right)$,

$$
\begin{aligned}
\mathrm{ISE}(\widehat{r}, r) \leq{}& 2\tau_0 \frac{\sum\limits_{i,j:\, y_i \neq y_j} \alpha_j K_\tau(\mathbf{x}_i - \mathbf{x}_j)}{n} - 2\tau_0 \frac{\sum\limits_{i,j:\, i \neq j, y_i = y_j} \alpha_j K_\tau(\mathbf{x}_i - \mathbf{x}_j)}{n-1} \\
&+ \tau_1 \sum_{y=1}^{2} \boldsymbol{\alpha}^{(y)^\top} (\mathbf{K}_{\sqrt{2}h}) \boldsymbol{\alpha}^{(y)} - \tau_1 \sum_{1 \leq i < j \leq n} 2\alpha_i \alpha_j K_{\sqrt{2}h}(\mathbf{x}_i - \mathbf{x}_j) \mathbb{I}_{y_i \neq y_j} + 2\tau_0 \varepsilon \\
\leq{}& 4\tau_0 \frac{\sum\limits_{1 \leq i < j \leq n} (\alpha_i + \alpha_j) K_\tau(\mathbf{x}_i - \mathbf{x}_j) \mathbb{I}_{y_i \neq y_j}}{n} - \tau_0 \frac{\sum\limits_{i,j=1}^{n} (\alpha_i + \alpha_j) K_\tau(\mathbf{x}_i - \mathbf{x}_j)}{n} \\
&+ \tau_1 \sum_{y=1}^{2} \boldsymbol{\alpha}^{(y)^\top} (\mathbf{K}_{\sqrt{2}h}) \boldsymbol{\alpha}^{(y)} - \tau_1 \sum_{1 \leq i < j \leq n} 2\alpha_i \alpha_j K_{\sqrt{2}h}(\mathbf{x}_i - \mathbf{x}_j) \mathbb{I}_{y_i \neq y_j} + 2\tau_0 \left(\frac{1}{n-1} + \varepsilon\right).
\end{aligned}
$$

$$(46)$$

The conclusion of this theorem can be obtained from (46). $\qquad\square$