# OpenReview forum: "Discriminative Similarity for Data Clustering"
_ICLR.cc/2022/Conference — ICLR 2022 Poster_

### Official Review · Reviewer_AedU · 2021-10-29

**Correctness:** 3
**Technical Novelty And Significance:** 3
**Empirical Novelty And Significance:** 3
**Recommendation:** 6
**Confidence:** 3

**Main Review:**

Strengths:
1. The paper is technically solid and it provides the theoretical analysis of the generalization bound.
2. The paper proposes a simple yet effective clustering method and provide the hyper-parameter tuning strategy.
3. The experimental results show the effectiveness of the proposed method.

Weaknesses:
1. The motivation is unclear for me. What are the benefits of the proposed discriminative similarity clustering method compared with other similarity based clustering methods?
2. The compared methods in Experiments are not the most recent ones. Among the 10 compared methods, only one was proposed in 2021, and all other methods were proposed before 2015. It would be better to compare with some more recent state-of-the-art clustering methods.
3. The computational complexity has been already analyzed in Page 6. I do not understand why  Section "Computational Complexity" is repeated in Experiments. Maybe the authors want to compare with other methods w.r.t. computational complexity? But I do not find such comparison. It would be better to show the comparison results of the  computational complexity and running time on the used data sets.
4. The Captions of Table 1 and Table 2 are exactly the same, which need to be corrected. Moreover, in the Caption, it says "$c$ in the left column is the cluster number", but I do not find $c$ in Tables 1 and 2.

**Summary Of The Paper:**

This paper proposes a discriminative similarity clustering method via unsupervised classification and provides the generalization bound for the similarity classification. The experimental results show the effectiveness of the proposed method.

**Summary Of The Review:**

Although there are some concerns, the paper proposes an interesting and effective method and also provides some theoretical analysis.  Therefore I recommend for weak accept.

---

> ### Author Response · Authors · 2021-11-26
> **Response to Reviewer AedU**
>
> We appreciate the review and the suggestions in this review.
>
> (1)  Please refer to (1) of Part 1 of our response to Reviewer qsFw for motivation and benefits of discriminative similarity compared with other similarity-based clustering methods.  We did have a dedicated section, the second last section of the original paper, for detailed discussion regarding the significance of CDSK over existing discriminative and similarity-based clustering methods. This section now becomes Section 2 of the revised paper. In (1) of Part 1 of our response to Reviewer qsFw, we also use a case study for subspace clustering (please refer to **Why the proposed discriminative similarity is better in separating clusters: a case study in subspace clustering**) to demonstrate the benefits of discriminative similarity when compared to the other sparse similarity learning method.
>
> (2) We appreciate the suggestion for including more recent baselines. We have performed a thorough literature review of existing discriminative and similarity-based clustering methods in Section 2 of the revised paper, and all the baseline methods with source code publicly available are included in our comparative experiments. While the source code of other baselines is not available, we still report comparison on commonly used dataset using the published results of the baselines with no source code. In particular, the discriminative Bayesian nonparametric clustering (Nguyen et al., 2017) achieves NMI of 0.62 on MNIST, while CDSK achieves NMI of 0.75 on the same dataset.
>
> (3) The computation complexity of CDSK is now only analyzed in a dedicated paragraph before Table 1 in the revised paper. The time complexity of different baselines and the running time on two representative datasets, Yale-B and MNIST, are shown in Table 3 in Section D.3 of the revised paper.
>
> (4) We have fixed the captions of Table 1 and Table 2 in the revised paper.

---

### Official Review · Reviewer_qsFw · 2021-11-01

**Correctness:** 4
**Technical Novelty And Significance:** 3
**Empirical Novelty And Significance:** 3
**Recommendation:** 6
**Confidence:** 4

**Main Review:**

Positive:
(1) Under the framework of CDS, discriminative similarity is induced by the generalization error bound for unsupervised similarity-based classifier. The authors conduct a complete and detailed theoretical analysis, and the results provide theoretical guarantee on the discriminative similarity can be induced from kernel density classification.
(2) Moreover, based on the CDS model, the authors develop a clustering algorithm termed Clustering by Discriminative Similarity via unsupervised Kernel classification (CDSK).
(3) CDSK uses a PSD kernel as the similarity function, and outperforms competing clustering algorithms, including nonparametric discriminative similarity based clustering methods and similarity graph based clustering methods, demonstrating the effectiveness of CDSK.

Negative:
(1)	Clustering performance highly depends on the effective data similarity. One of the main contributions of this paper is to propose a discriminative similarity. As is known to all, there exists a lot of classic similarity learning paradigms, such as metric learning methods and subspace learning methods. What are the main differences between these methods and the proposed method in this paper. I hope that the authors will theoretically or experimentally discuss the specific similarities and differences in a more detail way. Taking LRR as an example, the authors hope to obtain the similarity matrix has a property of low rank. What kind of properties does the discriminative similarity contain in this paper.
(2)	The authors let S^K_ij=2(\alpha_i+\alpha_j-\lambda\alpha_i\alpha_j)K(x_i-x_j) be the discriminative similarity between data from different classes. The authors did not clearly explain what the essential definition of this formula is. Could the authors explain the physical meaning of this formula more specifically. I hope the author can give an example to explain how does the similarity with learnable weights reflect discrimination.
(3)	The authors use Gaussian kernel as the predefined kernel, whether different predefined kernel have an impact on the final clustering results.
(4)	Does there exist kernel learning methods based on samples weighting, I believe there may be exist. It is suggested that the authors make a deep analysis on the related work and summarize what are the key differences between these efforts and the proposed method in this paper.
(5)	The authors conduct a complete and detailed theoretical analysis, but the description of the algorithm process is not clear and the optimization process is not detailed enough.
(6)	The experiments are inadequate in this paper, it is hoped that the authors give the parameter analysis and convergence analysis of the algorithm in an experimental way. The authors should compare the efficiency of different algorithms through the running time of different algorithms.


**Summary Of The Paper:**

This paper proposes a new clustering framework called Clustering by Discriminative Similarity (CDS). CDS learns an unsupervised similarity-based classifier from each data partition, and searches for the optimal partition of the data by minimizing the generalization error of the learnt classifiers associated with the data partitions. In contrasts with kernel similarity with uniform weights, the induced discriminative similarity with learnable weights enhances its capability to represent complex inter-connection between data. Based on CDS, CDSK is proposed as a new clustering method, with its effectiveness demonstrated by experimental results.

**Summary Of The Review:**

The paper contributes some new ideas and the motivation of this paper is clear. This paper proposes a new clustering framework termed Clustering by Discriminative Similarity (CDS). Based on this model, the authors develop a CDSK clustering algorithm. The paper is well organized. The authors conduct a complete and detailed theoretical analysis. Experiments on real-world datasets validated the effectiveness of the proposed methods.

---

> ### Author Response · Authors · 2021-11-26
> **Response to Reviewer qsFw Part 1**
>
> We appreciate the review and the suggestions in this review.
>
> (1) **Difference from Existing Similarity Learning Methods and Our Significance**
>
> Thank you for asking for the difference between our method and other similarity learning methods. The second last section of the original paper is dedicated to a detailed discussion about such difference and the significance of our method. This section now becomes Section 2 of the revised paper, which is presented below for your convenience. The references are included in the last part of the response series.
>
> Effective data similarity highly depends on the underlying probabilistic distribution and geometric structure of the data, and these two characteristics leads to “data-driven” similarity, such as ref. [D,E,F,M,N,O,P], and similarity based on geometric structure of the data, such as the subspace structure (Sparse Subspace Clustering, or SSC in ref. [I]). Note that the sparse graph method, $\ell^1$-Graph (ref. [R]), has the same formulation as SSC. Most existing clustering methods based on data-driven or geometric structure-driven similarity suffer from a common deficiency, that is, the similarity is not explicitly optimized for the purpose of separating underlying clusters. In particular, the RF (Random Forest)-based similarity (ref. [D,F]) is extracted from features in decision trees. Previous works about subspace-based similarity (ref. [R] and ref. [I]) try to make sure that only data points lying on or close to the same subspace have nonzero similarity, so that data points from the same subspace can form a cluster. However, it is not guaranteed that features in the decision trees are discriminative enough to separate clusters, because the candidate data partition (or candidate cluster labels) do not participate in the feature/similarity extraction process. Note that synthetically generated negative class are suggested in ref. [D,F] to train unsupervised random forest, however, the synthetic labels are not for the original data. Moreover, it is well known that the existing subspace learning methods only obtain reliable subspace-based similarity with restrictive geometric assumptions on the data and the underlying subspaces, such as large principal angle between intersecting subspaces (ref. [I,J]). In addition, the data-dependent similarity measures in ref. [E] do not require a learning process, so they are not optimized for separating clusters. In fact, the similarity measures in [E] are evaluated for the information retrieval and kNN classification tasks.
>
> Therefore, it is particularly important to derive similarity for clustering which meets two requirements: (1) discriminative measure with information such as cluster partition is used to derive such similarity so as to achieve compelling clustering performance; (2) it requires less restrictive assumptions on the geometric structure of the data than current geometric structure-based similarity learning methods, such as subspace clustering (ref. [R] and ref. [I]).
>
> **Significance**  The proposed discriminative similarity meets these two requirements. First, the discriminative similarity is derived by the generalization error bound associated with candidate cluster labeling, and minimizing the objective function of our optimization problem for clustering renders a joint optimization of discriminative similarity and candidate cluster labeling in a way such that the similarity-based classifier has small generalization error bound. Second, our framework only assumes a mild classification model in Definition 3.1, which only requires an unknown joint distribution over data and its labels. In this way, the restrictive geometric assumptions are avoided in our method. Compared to the existing discriminative clustering methods, such as MMC (ref. [Q]), BMMC (ref. [B]), RIM (ref. [S]), and the two other discriminative clustering methods (ref. [A, C]), the optimization problem of CDSK with discriminative similarity-based formulation is much easier to solve and it enjoys convexity and efficiency in each iteration of coordinate descent described in Algorithm 1. In particular, as mentioned in Section D of the supplementary, the first step (eq. [11]) of each iteration can be solved by efficient SVD or other randomized large-scale SVD methods, and the second step (eq. [12]) of each iteration can be solved by efficient SMO (ref. [T]). Moreover, the optimization problems in these two steps are either convex or having closed-form solution. In contrast, MMC requires expensive semidefinite programming. RIM has to solve a nonconvex optimization problem and its formulation does not guarantee that the trained multi-class kernelized logistic regression has low classification error on candidate labeling, which explains why it has inferior performance compared to our method.

---

> > ### Author Response · Authors · 2021-11-26
> > **Response to Reviewer qsFw Part 2**
> >
> > **Significance** (Cont'd) The discriminative Bayesian nonparametric clustering (ref. [C]) and BMMC (ref. [B]) require extra efforts of sampling hidden variables and tuning hyperparameters to generate the desirable number of clusters (or model selection), which could reduce the effect of discriminative measures used in these Bayesian nonparametric methods.
> >
> > Below is an example explaining why discriminative similarity is better than the existing sparse similarity in subspace clustering and the geometric property that discriminative similarity possesses, that is, encouraging nonzero discriminative similarity only between data on the same subspace so that the subspace structure of the data is preserved.
> >
> > **Why the proposed discriminative similarity is better in separating clusters: a case study in subspace clustering**
> >
> > We explain why the proposed discriminative similarity achieves superior performance for subspace clustering, when compared to other Sparse Subspace Clustering (SSC) methods which learn sparse similarity between data. Given a set of data points lying on or close to a union of subspaces, subspace clustering methods aim to identify data points belonging to the same subspace as a cluster. The primary goal of similarity-based subspace clustering is to obtain a similarity measure such that data points belonging to the same subspace has nonzero similarity and data points from different subspaces have zero similarity. In this way, clustering based on such similarity respecting the underlying subspace structure of the data is expected to have compelling performance. We adopt a measure called Relative Violation (RV), which is also used in ref. [K], to measure the quality of the proposed similarity and the sparse similarity learned by established SSC methods (ref. [R] and ref. [I]). Given a pairwise similarity matrix, RV evaluates the ratio of similarity between data from different subspaces (clusters) to that for data from the subspaces. It can be seen that the similarity with a smaller RV indicates that it more faithfully reflects the subspace structure of the data. It is known that face images belonging to the same person with fixed pose and illumination conditions approximately lie on or close to the same subspace (ref. [I]), so we use face datasets to demonstrate the RV values for subspace clustering. It is shown in the table below that discriminative similarity renders much lower RV than that of ref. [R] and ref. [I], explaining why CDSK based on discriminative similarity has better performance for subspace clustering.
> >
> > | RV\Datasets                                           | Yale-B    | Georgia Face  | CMU PIE | UMIST Face | MPIE S1  | MPIE S2 | MPIE S3  | MPIE S4 |
> >  | ----------------------------------------------------| ------------| ------------------  | -------------| ------------------|--------------| -----------|--------------| -----------|
> >  | Sparse Similarity (ref. [R] and ref. [I])  |0.26         |1.38                  | 0.15        |0.13               | 0.16           |0.14  |  0.14      |0.14             |
> >  | Discriminatively Similarity                     |0.13         |0.48                  | 0.07        |0.06               | 0.25            |0.21  | 0.21      | 0.22            |
> >
> > (2) Physical Meaning of Discriminative Similarity
> >
> > Section D.1 of the revised paper explains the physical meaning of discriminative similarity, in particular, the physical meaning of adaptive weights in discriminative similarity with illustration in Figure 2.
> >
> > (3) Our theoretical results apply to any continuous and symmetric similarity function $S \colon \mathcal X \times \mathcal X \to [0,1]$ (where the image of this function, $[0,1]$, can be generalized to $[0,a]$ for arbitrary positive $a > 0$) which satisfies the conditions specified in Lemma 4.1. In particular, because every PSD kernel satisfies the conditions in Lemma 4.1, it follows that our main theoretical result, Theorem 4.2, applies to every PSD kernel, including Gaussian kernel and other predefined PSD kernel. This paper focuses on using Gaussian kernel in the discriminative similarity, and it remains an interesting empirical investigation in the future about the performance of discriminative similarity using other PSD kernels.

---

> > > ### Author Response · Authors · 2021-11-26
> > > **Response to Reviewer qsFw Part 3**
> > >
> > > (4) Thank you for pointing out kernel learning methods based on samples weighting. As discussed in the main paper, the similarity learning method proposed by Balcan et al. in 2008 uses similarity-based classifier with sample weights in the form of eq. [1], and the similarity function $S$ can be a kernel function.  In addition, the well-known Support Vector Machines and Multiple Kernel Learning (Lanckriet et al. Learning the kernel matrix with semidefinite programming. Journal of Machine Learning Research, 2004) are kernel learning methods with samples weighting. It is worthwhile to mention that most kernel learning methods based on samples weighting are not for unsupervised learning tasks. To the best of our knowledge, our proposed method is among the very few methods in the literature using samples weighting for unsupervised learning, with theoretical guarantee on the generalization error bound for the corresponding unsupervised classifier in Theorem 4.2. Notably, our general result in Theorem 4.2 reduces to the well-established generalization error bound for Support Vector Machines or Kernel Machines when the similarity function is a PSD kernel. We will discuss these details about kernel learning methods based on samples weighting in the final version of this paper.
> > >
> > > (5) Section D.4 of the revised paper describes more details about the optimization process of CDSK. In particular, each subproblem in the iteration of coordinate descent in Algorithm 1 is explained, and more details about Sequential Minimal Optimization (SMO) used for the second subproblem are included.
> > >
> > > (6) **Parameter Analysis:** the optimization process of CDSK optimizes two parameters, $\mathbf Y$ and $\mathbf \alpha$. $\mathbf Y \in \mathbb R^{n \times c}$ has $n \times c$ individual parameters, and $\mathbf \alpha \in \mathbb R^{n}$ has $n$  parameters, where $n$ is the total number of data points and $c$ is the number of clusters.
> > >
> > > **Empirical Convergence Analysis** The value of the objective function, that is, $Q$ in equation (9'') in Section C of the appendix, with respect to the iteration number of coordinate descent on the Yale-B dataset is illustrated in Figure 3 of the revised paper. This shows the empirical convergence of the objective function for CDSK.
> > >
> > > The time complexity of different baselines and the running time (in seconds) on two representative datasets, Yale-B and MNIST, are shown in Table 3 in Section D.3 of the revised paper. Moreover, we report more comparative result in part (2) of our response to Reviewer AedU.

---

> > > > ### Author Response · Authors · 2021-11-26
> > > > **Response to Reviewer qsFw Part 4 (References)**
> > > >
> > > > **References:**
> > > > - [A] Gao Huang, Tianchi Liu, Yan Yang, Zhiping Lin, Shiji Song, and Cheng Wu. 2015. Discriminative clustering via extreme learning machine. Neural Netw. 70, C (October 2015), 1–8.
> > > > - [B] Changyou Chen, Jun Zhu, and Xinhua Zhang. 2014. Robust Bayesian max-margin clustering. In Proceedings of the 27th International Conference on Neural Information Processing Systems - Volume 1 (NIPS'14), 532–540.
> > > > - [C] Vu Nguyen, Dinh Phung, Trung Le, and Hung Bui. 2017. Discriminative bayesian nonparametric clustering. In Proceedings of the 26th International Joint Conference on Artificial Intelligence (IJCAI'17). AAAI Press, 2550–2556.
> > > > - [D] Xiatian Zhu, Chen Change Loy, Shaogang Gong. Constructing Robust Affinity Graphs for Spectral Clustering. CVPR 2014: 1450-1457
> > > > - [E] Aryal, S, Ting, KM, Washio, T & Haffari, G 2020. A comparative study of data-dependent approaches without learning in measuring similarities of data objects, Data Mining and Knowledge Discovery, vol. 34, no. 1, pp. 124-162.
> > > > - [F] M. Bicego, F. Cicalese and A. Mensi, "RatioRF: a novel measure for Random Forest clustering based on the Tversky's Ratio model," in IEEE Transactions on Knowledge and Data Engineering (TKDE), 2021.
> > > > -[G] L. Xu and D. Schuurmans. Unsupervised and semi-supervised multiclass support vector machines. Proc. 20th Nat. Conf. Artif. Intell., 2005, pp. 904–910.
> > > > -[H] Oriol Vinyals, Charles Blundell, Timothy Lillicrap, Koray Kavukcuoglu, and Daan Wierstra. 2016. Matching networks for one shot learning. In Proceedings of the 30th International Conference on Neural Information Processing Systems (NIPS'16), 3637–3645.
> > > > -[I] Elhamifar, E., Vidal, R.: Sparse subspace clustering: Algorithm, theory, and applications. IEEE Trans. Pattern Anal. Mach. Intell. 35(11) (2013) 2765-2781.
> > > > -[J] Mahdi Soltanolkotabi and Emmanuel J. Cands. A geometric analysis of subspace clustering with outliers. Ann. Statist., 40(4):2195{2238, 08 2012.
> > > > -[K] Yu-Xiang Wang and Huan Xu. Noisy sparse subspace clustering. In Proceedings of the 30th International Conference on Machine Learning, ICML 2013, pages 89-97, 2013.
> > > > -[L] Julien Mairal, Francis Bach, and Jean Ponce. 2012. Task-Driven Dictionary Learning. IEEE Trans. Pattern Anal. Mach. Intell. 34, 4 (April 2012), 791–804.
> > > > -[M] Andrew Y. Ng, Michael I. Jordan, and Yair Weiss. On spectral clustering: Analysis and an algorithm. In NIPS, pages 849–856, 2001.
> > > > -[N] Noam Shental, Assaf Zomet, Tomer Hertz, and Yair Weiss. Pairwise clustering and graphical models. In NIPS, 2003.
> > > > -[O] J. A. Hartigan andM. A.Wong. A K-means clustering algorithm. Applied Statistics, 28:100–108, 327 1979.
> > > > -[P] Bernhard Schölkopf, Alexander Smola, and Klaus-RobertMüller. Nonlinear component analysis as a kernel eigenvalue problem. Neural Comput., 10(5):1299–1319, July 1998.
> > > > -[Q] Linli Xu, James Neufeld, Bryce Larson, and Dale Schuurmans. Maximum margin clustering. In NIPS, 2004.
> > > > -[R] Shuicheng Yan and Huan Wang. Semi-supervised learning by sparse representation. In SDM, pages 792–801, 2009.
> > > > -[S] Ryan Gomes, Andreas Krause, and Pietro Perona. Discriminative clustering by regularized information maximization. In NIPS, pages 775–783, 2010.
> > > > -[T] John Platt. Sequential minimal optimization: A fast algorithm for training support vector machines. Technical report, 1998.

---

### Official Review · Reviewer_dtfu · 2021-11-02

**Correctness:** 4
**Technical Novelty And Significance:** 4
**Empirical Novelty And Significance:** 4
**Recommendation:** 8
**Confidence:** 5

**Main Review:**

Strengths:

The proposed CDS framework seems interesting and it is a principled framework of data clustering. In fact, the maximum margin based clustering methods are implicitly based on such a framework, and it seems nice to explicitly formulate this framework in this paper. This paper also provides strong theoretical results for the generalization error bound of unsupervised similarity-based classifier using Rademacher complexity. As explained in Remark 3.4 and the appendix, the derived generalization bound is a generalized version of the well-established generalization bound for kernel machines, so it has an independent theoretical interest.

This paper also conducts extensive experimental results with various baselines from the similarity-based clustering and discriminative clustering literature. I particularly appreciate the detailed discussion in Section 5, which places CDSK in a clear position in the literature and explains its significance.

Weakness:

It could add more value to this paper if the authors provide more examples showing why the derived discriminative similarity is better than conventional similarities, such as the regular kernel similarity and the similarity used in the sparse graph (especially subspace)-based clustering literature.


**Summary Of The Paper:**

This paper proposes a new discriminative clustering framework by learning an unsupervised classifier from unlabeled data. An unsupervised classifier is learned from every hypothetical data partition, and the optimal data partition is then regarded as the partition corresponding to the minimum generalization error bound of the unsupervised classifier. The authors use a similarity-based classifier as the unsupervised classifier in this framework and derive its generalization error bound as the sum of discriminative similarity between different clusters. The proposed algorithm, CDS via unsupervised kernel classification (CDSK), then minimizes the between-cluster discriminative similarity. Extensive experimental results with the comparison with a broad range of baselines confirm the superior performance of the proposed algorithm.

**Summary Of The Review:**

This paper presents a novel and interesting clustering method with theoretical explanations, and the detailed discussion with the similarity-based clustering and discriminative clustering literature is appreciated. I encourage the authors to add more examples showing the comparison between the derived discriminative similarity and conventional similarity used in data clustering.

---

> ### Author Response · Authors · 2021-11-26
> **Response to Reviewer dtfu**
>
> We appreciate the review and the suggestions in this review.
>
> Please refer to Part 1 of our response to Reviewer qsFw and Section 2 of the revised paper regarding the significance of CDSK over existing discriminative and similarity-based clustering Methods. In particular, In (1) of Part 1 of our response to Reviewer qsFw, we use a case study for subspace clustering (please refer to **Why the proposed discriminative similarity is better in separating clusters: a case study in subspace clustering**) to demonstrate the advantage of discriminative similarity when compared to the other sparse similarity learning method.

---

### Official Review · Reviewer_TUzr · 2021-11-03

**Correctness:** 2
**Technical Novelty And Significance:** 2
**Empirical Novelty And Significance:** 2
**Recommendation:** 5
**Confidence:** 3

**Main Review:**

Strength:
This paper presents a learnable similarity function that adaptively adjusts the weights of the similarity between one sample and the samples of some class.
The authors also provide theoretical results to support the propose algorithm and give the strict proofs.
The experimental results of the proposed algorithm outperform that of the other comparable clustering algorithms.

Weaknesses:
1. The expected loss in Eq. (4) seems to be not matched with the corresponding empirical loss. Moreover, the authors should explain how the empirical loss in Eq. (4) can be transformed into Eq. (5). The detailed derivative process should be listed in the appendix. This is important.
2. It’s not suitable to use boldface to represent a scalar variable.
3. There are some typos in this paper.
4. Some variables lack necessary description, e.g., \mathbb{1}_{y_i \neq y_j} in Eq. (7).


**Summary Of The Paper:**

This paper proposes a clustering algorithm by learning from the weighted similarity of each data partition. More precisely, the authors propose an unsupervised framework which aims to minimize an objective of a similarity-based classifier. Then, the upper bound of excess risk of this classifier is provided by Rademacher complexity. They show that the weighted kernel similarity is a special case of the general framework, and they apply it to discriminative clustering by iteratively optimizing the class labels and the weights of similarity. Finally, experiments are conducted to verify the effectiveness of the proposed algorithm.

**Summary Of The Review:**

The details of the paper needs further refinement.

---

### Decision · Program_Chairs · 2022-01-20

**Decision:**

Accept (Poster)

**Comment:**

The authors provide a framework for unsupervised clarification based on minimizing a between-cluster discriminative similarity. It is more flexible than existing methods whose kernel similarity implicitly assumes uniform weights, and the authors connect to ideas such as max-margin and weighted kernel approaches. This yields a clustering algorithm naturally that alternates between updating class labels and similarity weights. Moreover the reviewers (and I) appreciate the analysis of generalization error through Rademacher complexity arguments and detailed author responses. I might add while the paper draws connections to weighted kernel methods and have since added references to sparse subspace clustering etc, there is recent interest in using similar arguments to derive error bounds and uniform concentration results for center-based methods that might be included in the survey of related work, for instance recent work from Swagatam Das and collaborators. The authors have importantly added details on the optimization using SMO, and the revision should include these details in a clear exposition together with the computational complexity discussion mentioned in their response.